# DATE: Dynamic Absolute Time Enhancement for Long Video Understanding

## Abstract

Long video understanding remains a fundamental challenge for multimodal large language models (MLLMs), particularly in tasks requiring precise temporal reasoning and event localization. Existing approaches typically adopt uniform frame sampling and rely on implicit position encodings to model temporal order. However, these methods struggle with long-range dependencies, leading to critical information loss and degraded temporal comprehension. In this paper, we propose **D**ynamic **A**bsolute **T**ime **E**nhancement **(DATE)** that enhances temporal awareness in MLLMs through the Timestamp Injection Mechanism (TIM) and a semantically guided Temporal-Aware Similarity Sampling (TASS) strategy. Specifically, we interleave video frame embeddings with textual timestamp tokens to construct a continuous temporal reference system. We further reformulate the video sampling problem as a vision-language retrieval task and introduce a two-stage algorithm to ensure both semantic relevance and temporal coverage: enriching each query into a descriptive caption to better align with the vision feature, and sampling key event with a similarity-driven temporally regularized greedy strategy. Our method achieves remarkable improvements w.r.t. absolute time understanding and key event localization, resulting in state-of-the-art performance among 7B and 72B models on hour-long video benchmarks. Particularly, our 7B model even exceeds many 72B models on some benchmarks.

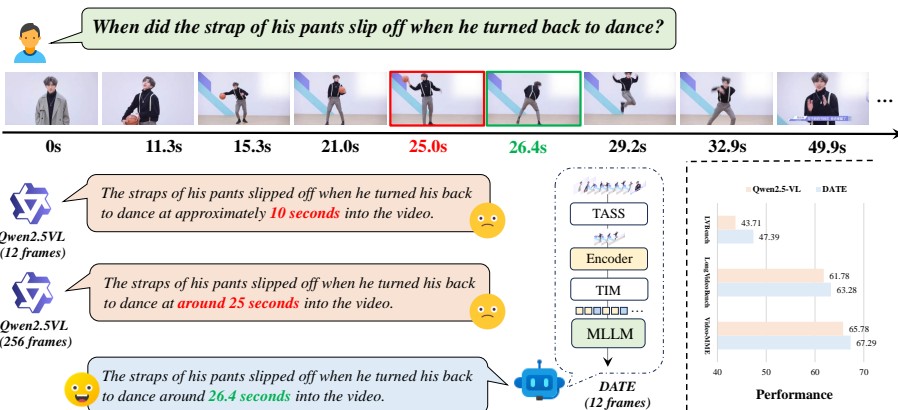

Figure 1: A **Real** example of our proposed DATE compared with Qwen2.5-VL. It shows DATE with 12 frames beats 256 frames of Qwen2.5-VL.

## 1 Introduction

Multimodal large language models (MLLMs)Alayrac et al. (2022); Cheng et al. (2024b); Wang et al. (2024a) have shown remarkable performance in a wide range of video understanding tasks, including video captioning, question answering, and event localization. However, when extended to long videos, these models face fundamental challenges in temporal reasoning and precise event localization. The essential reason for this limitation lies in the mismatch between rigid input length constraints of

transformer architectures and the inherently long and continuous nature of real-world video content. As a result, existing approaches typically resort to uniform frame sampling as a preprocessing step. Unfortunately, this coarse-grained strategy often leads to the loss of critical visual events, temporal discontinuity, and the collapse of causality chains, severely limiting the model's capacity to reason over spatiotemporal structures. Moreover, there is no ability to perform perception and alignment of the absolute time and the corresponding frames.

One major obstacle is the inability of current methods to construct explicit representations of **absolute time**. Even when time-stamped subtitles are used as prompts, models struggle to align absolute timestamps with specific video frames. Although models such as Qwen2.5VLBai et al. (2025) incorporate absolute time information into the temporal position embedding based on Multimodal RoPEWang et al. (2024a); Su et al. (2024), this approach exhibits critical drawbacks: For short video clips, time differences within one second remain indistinguishable; for long videos, the continual growth of positional indices leads to a loss of relative positional perception and eventual degradation of temporal comprehension. Our diagnostic experiments further confirm that such models do not solve problems related to absolute time reliably.

Another significant challenge comes from frame sampling itself. Uniform discretizations of frames lead to sparse observations, especially in long videos where adjacent frames may be separated by tens of seconds. Such sampling is agnostic to semantic content and fails to adapt dynamically to user queries, resulting in low recall when critical events are temporally sparse. Recent methods like Adaptive Keyframe Selection (AKS)Tang et al. (2025) attempt to mitigate this by introducing query-guided dynamic sampling. However, they suffer from two key issues: (1) they use raw user questions as CLIPRadford et al. (2021) text encoders, which contradicts CLIP's training paradigm centered on descriptive captions, leading to unstable or truncated representations; (2) their sampling method may still select irrelevant frames (e.g., negative samples with relatively high scores) and often fails in visually stable segments due to insufficient score variance.

To address these limitations, we proposed DATE, as shown in Fig.2, for absolute time-aware video understanding and event localization. Our method builds a temporal coordinate system directly within the multimodal sequence by interleaving explicit timestamp tokens with video frame embeddings. This timestamp injection preserves visual continuity while allowing for precise and controllable temporal references. To guide the model towards relevant content, we formulate video sampling as a text-image retrieval task and employ a two-stage semantic-guided selection strategy: (i) rewriting user questions into caption-style descriptions for better alignment with CLIP-based vision-language similarity computation, and (ii) applying a temporally-regularized greedy sampling algorithm that ensures both high semantic relevance and temporal diversity. Our contributions are three-folds:

(1) We introduce **Timestamp Injection Mechanism (TIM)** that enables explicit absolute time modeling without modifying model weights or requiring additional training.

(2) We propose **Temporally-Aware Similarity Sampling (TASS)**, a temporally-regularized greedy sampling algorithm with semantic-guided caption generation to sample frames, which balance key events with video continuity.

(3) We show that our method achieves superior **spatial perception** and **event localization**, especially for **hour-long** video scenarios, which achieve SOTA on 7B models, even surpassing many 72B models. Moreover, the DATE-72B model achieves state-of-the-art performance.

## 2 RELATED WORKS

### 2.1 MULTIMODAL LARGE LANGUAGE MODELS FOR VIDEO UNDERSTANDING

With the widespread success of large language models (LLMs) Achiam et al. (2023); Brown et al. (2020); Chiang et al. (2023); Chowdhery et al. (2023); Chung et al. (2024); Grattafiori et al. (2024); Touvron et al. (2023a;b); Ray (2023); Chen et al. (2024c) in natural language processing, researchers have extended these models to multimodal scenarios, forming multimodal large language models (MLLMs)Lai et al. (2024); Liu et al. (2023). By incorporating visual encoders, MLLMs are capable of processing visual inputs such as images or videos, enabling tasks like visual question answering, video captioning, and visual reasoningMaaz et al. (2023); Alayrac et al. (2022); Chen et al. (2024a); Wu et al. (2024a); Min et al. (2024); Qian et al. (2024); Wang et al. (2022). Representative models

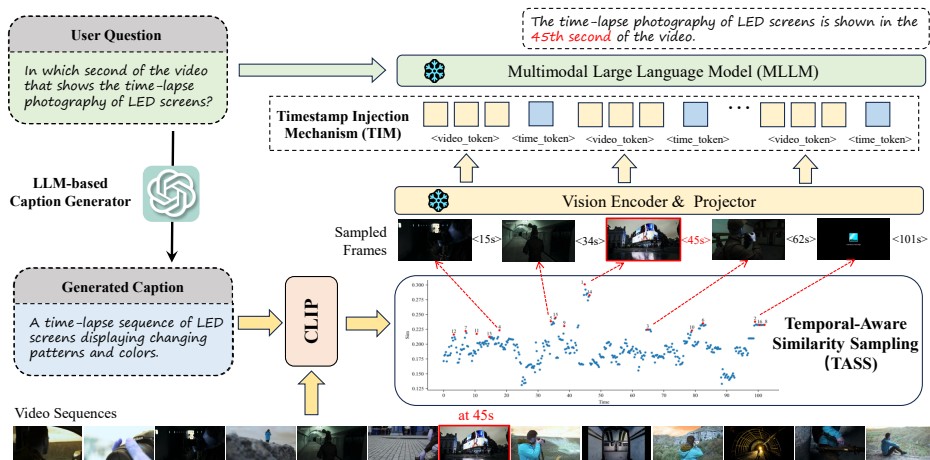

Figure 2: Overview of the proposed framework. For each user input question, using LLM-based Caption Generator to generate a CLIP-aligned image caption, and calculate the similarity with video frames. Then, use Temporal-Aware Similarity Sampling (TASS) strategy to sample the frames (The real sampled frames and orders of this demo could be found in **Appendix B**). Last, with Timestamp Injection Mechanism (TIM), we embed timestamps aligned with each frame.

include Video-ChatGPTMaaz et al. (2023); Lin et al. (2023), LLaVA-VideoZhang et al. (2024b), VideoLLAMAZhang et al. (2023); Cheng et al. (2024b); Zhang et al. (2025), and Qwen-VLWang et al. (2024a); Bai et al. (2025), which typically encode video frames into visual tokens and feed them into the model alongside textual tokens. However, due to the inherent context length limitations of LLMs, these models often rely on fixed frame sampling strategies, resulting in significant information compression when processing long video dataFu et al. (2024); Wu et al. (2024b); Wang et al. (2024b). Moreover, long videos present unique challenges such as sparse events and wide semantic spans, which demand more effective temporal modeling and cross-segment reasoning capabilities. Therefore, many strategies Shang et al. (2024); Zhang et al. (2024a); Wei & Chen (2024); Chen et al. (2024d); Wang et al. (2025); Cheng et al. (2024a); He et al. (2024b;a) proposed for longer context.

## 2.2 TEMPORAL MODELING

Temporal modeling is a fundamental challenge in long video understanding. Existing methods can be broadly categorized into two groups: ①Using data with timestamps to fine-tune model with time tokensChen et al. (2024b) or prompts with timestampsRen et al. (2024). These need more data and training cost. ②Explicit incorporation of time into positional encoding. For example, Qwen2.5VL introduces MRoPEBai et al. (2025) and Qwen2.5-OmniXu et al. (2025) introduces TMRoPE, which use absolute time signals into its rotary positional encoding. However, this encoding mechanism is prone to positional drift in long sequences, where the encoded position values grow too quickly with sequence length, thereby distorting the relative temporal relationships between frames. This can reduce the ability of the model to capture temporal causality and duration. More importantly, these methods often fail to provide a stable temporal awareness, thus limiting the ability of the model to perceive absolute time.

## 2.3 FRAME SAMPLING STRATEGY

To mitigate the performance bottleneck caused by limited input length, frame sampling has become a crucial component in video understanding systems. The most common strategy is uniform samplingBai et al. (2025); Cheng et al. (2024b); Li et al. (2024), which is straightforward but fails to adaptively select frames based on semantic importance. This often leads to omission of critical content, especially in videos with dense or uneven event distributions. To address this, some semantics-aware frame selection methods with VLMs like CLIPRadford et al. (2021) have been proposed, such as BOLTLiu et al. (2025) and AKSTang et al. (2025), and they proved to be effective over uniform and topk sampling. However, they all use question to find frames, this is not a good

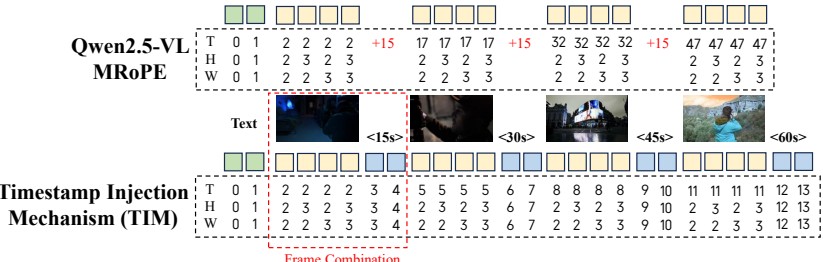

Figure 3: The Multimodal RoPE (MRoPE) with our Timestamp Injection Mechanism (TIM) compared with Qwen2.5-VL's MRoPE. **Qwen2.5-VL:** Add 15 since there are 15 seconds betweet frames. **TIM(ours):** The temporal dimension $T$ is extended with time token. The spatial dimensions $(H, W)$ remain aligned with the first frame, ensuring spatial consistency across the whole sequence.

way for CLIP to embed question, since it was not trained with question. Meanwhile, they may also sample negative frames and loss critical temporal continuity (action, movement, etc.).

## 3 METHODS

### 3.1 TIMESTAMP INJECTION MECHANISM (TIM)

To enhance the temporal perception of Multimodal Large Language Models (MLLMs) in video understanding, especially in long videos requiring absolute time localization, we propose a timestamp injection mechanism. This mechanism is model-agnostic and compatible with most mainstream MLLMs. In this work, we take Qwen2.5-VLBai et al. (2025), which incorporates explicit absolute time encoding, as our baseline method.

**Token-Level Timestamp Injection**    The latest open-source MLLM, Qwen2.5-VL, relies on their proposed MRoPE (Multimodal RoPE) mechanism to model temporal sequences with time interval in the position ID of MRoPEWang et al. (2024a), to embed absolute time of video frames. However, our experiments demonstrate that this approach lacks a true understanding of absolute time.

To address this, we introduce a token-level timestamp injection mechanism. As shown in Fig.3, for each sampled frame, we construct the input sequence using an interleaved structure of visual and time tokens:

<video_token><time_token><video_token><time_token> ...<video_token><time_token>

Here, each color represents the combination of video tokens and timestamps of a frame, <video_token> represents the visual tokens (not one token), and <time_token> is its corresponding textual timestamp (e.g., 01:23 or 83s). This structure preserves visual continuity while injecting a precise and controllable temporal reference, enabling the language model to perform time-aware reasoning task such as event ordering and absolute time localization.

**Reconstruction of Positional Encoding and Sequential Normalization**    The MRoPE mechanism in Qwen2.5-VL introduces absolute time information via position indices in the visual branch. Although it models temporal order to some extent, it suffers from critical limitations when applied to long videos due to linearly increasing position indices(IDs):

**(1) Sparsity and Resource Inefficiency:** Since position IDs grow proportionally, large time gaps (e.g., 20s between frames) leading to inefficient use of the sequence length and potential index explosion (e.g., 10,000 in hour-long videos).

**(2) Degradation of Relative Positional Awareness:** Large gaps between position IDs disrupt the relative distances between tokens, compromising the ability to capture local temporal structures.

To mitigate these issues, we remove the absolute time alingment from Qwen2.5VL's MRoPE and retain only the original Multimodal RoPE (MRoPE) encoding. Specifically, the temporal dimension $T$

is encoded using a simple *sequential indexing* strategy, where position indices increment according to the order of tokens. Furthermore, to preserve the spatial encodings between video frames, we ensure that only the temporal dimension $T$ is extended along with time token insertion. The spatial encodings $(H, W)$ remain aligned with the first frame, ensuring spatial consistency across the sequence.

This design maintains the numerical stability of RoPESu et al. (2024), and preserves the model's sensitivity to token order. Meanwhile, absolute time perception is handled independently via the explicit `<time_token>`s, resulting in a decoupled and robust time representation framework. Moreover, as shown in Fig.6, a modality gap between vision tokens and time tokens makes the model can better locate them key events. As the result of the proof in AppendixB.2, when position encoding for each frame is less than 6.28, it could perceive relative positions better. Therefore, for ours TIM, the video tokens use one position id, and the time token use less than four position ids, which uses a total less than five position ids for each frame.

### 3.2 TEMPORAL-AWARE SIMILARITY SAMPLING (TASS)

Discretized video frame sampling is a common preprocessing step in multimodal video modeling. However, in long video scenarios, uniformly spaced sampling strategies exhibit clear limitations. On the one hand, the temporal gaps between frames may span several seconds to minutes, making it likely to miss sparse but semantically critical moments. On the other hand, uniform sampling is task-agnostic, severely undermining the recall of key events.

Sampling directly based on similarity leads to frames with little variation being sampled continuously, which results in video features collapsing into a single image. Sampling across too large a span would then lead to problems with key event continuity, difficulty in recognizing object movement, etc., i.e., a similar problem to that which would occur with uniform sampling and AKSTang et al. (2025).

Thus, we proposed **TASS**, a temporally-regularized greedy sampling algorithm that ensures both high key event continues and temporal diversity. It consists of two main stages: (i) *semantic-enhanced similarity computation*, and (ii) *similarity-prioritized sampling under temporal constraints*.

**Semantic Enhancement: From Question to Caption**   To improve the consistency of the visual-language alignment, we first convert the user's query (typically a question) into a more descriptive caption using a language model, and the prompt of this step can be seen in Appendix H. Unlike raw questions, captions exhibit a declarative style that aligns better with CLIP's image-text matching paradigm, activating more stable and complete semantic representations.

Each video frame $v_i$ is embedded using CLIP, and its similarity to the caption $c$ is calculated as:

$$s_i = \text{CLIP}(v_i, c) = \frac{\langle v_i, c \rangle}{\|v_i\| \cdot \|c\|} \tag{1}$$

**Temporal-Aware Similarity Sampling**   We first compute a dynamic threshold $s_{\text{mean}}$ which is the mean of all similarity scores. Scores below the mean are considered *negative samples*, as they contribute little to answering the user's query and are therefore discarded. To ensure computational efficiency, we further cap the number of top-ranked candidates by setting an upper bound proportional to the final number of selected frames, i.e., $\text{topk} \leq 4 \times \text{max\_frames}$.

$$\text{topk} = \min \left( |\{i \mid s_i > s_{\text{mean}}\}|, \, \alpha \times \text{max\_frames} \right) \tag{2}$$

where $\alpha$ is a controllable coefficient. It denotes the number of frames to be sampled (candidate frames). For example, Qwen2.5-VL-7B can process up to 256 frames, and we set $\alpha = 4$ by default, using our sampling strategy, we can effectively compress and select representative frames from a sequence of $4 * 256 = 1024$ frames. When negative sample filtering is considered, the expected number of candidate frames for sampling could be 2048.

While many continuous frames are semantically aligned, they often cluster temporally, leading to redundancy. To ensure temporal diversity while preserving semantic relevance, we introduce a greedy selection algorithm that is similarity first with enforcing a minimum time interval $\delta$ between selected timestamps. If fewer than $N_{\text{max}}$ frames are obtained, $\delta$ is iteratively decayed until the quota is met. The pseudo-code is as follows:

---

**Algorithm 1** Temporal-Aware Similarity Sampling (TASS)

---

**Require:** Top-K timestamps $\mathcal{I}_{\text{topK}}$, sampled frames $N_{\max}$, initial interval $\delta_0$
**Ensure:** Selected timestamps $\mathcal{S}_t$
1: Initialize $\mathcal{S}_t \leftarrow \emptyset$, $\delta \leftarrow \delta_0$, decay ratio $\lambda = 0.5$
2: **while** $|\mathcal{S}_t| < N_{\max}$ **do**
3:     **for** each $t_k \in \mathcal{I}_{\text{topK}}$ **do**
4:         **if** $\forall t_j \in \mathcal{S}_t, |t_k - t_j| \geq \delta$ or $\mathcal{S}_t = \emptyset$ **then**
5:             $\mathcal{S}_t \leftarrow \mathcal{S}_t \cup \{t_k\}$
6:             Remove $t_k$ from $\mathcal{I}_{\text{topK}}$
7:             **if** $|\mathcal{S}_t| \geq N_{\max}$ **then**
8:                 **break**
9:             **end if**
10:         **end if**
11:     **end for**
12:     $\delta \leftarrow \delta \cdot \lambda$
13: **end while**
14: **return** sorted $\mathcal{S}_t$

---

The most relevant work w.r.t. TASS is the Adaptive Keyframe Selection (AKS) proposed by Tang et al.Tang et al. (2025), which introduces a query-driven sampling mechanism. However, it suffers from two major issues: (1) It directly uses raw questions as CLIP text inputs, misaligned with CLIP's caption-style since it was trained with image-caption pairs but not questions, and prone to semantic truncation due to the input limitation; (2) Its variance-based sampling strategy tends to include false positives (i.e., high-scoring frames from negative segments), due to the small magnitude of score variations, and may miss keyframes in visually smooth regions.

In contrast, our method leverages caption rewriting for better alignment and introduces a temporal regularization mechanism to ensure broader temporal coverage. This makes sampling more robust and effective for modeling temporally distributed events in long videos.

## 4 EXPERIMENTS

### 4.1 BENCHMARKS

To comprehensively evaluate our proposed DATE on long video understanding, we conduct experiments on three hour-long video benchmarks that emphasize complex temporal reasoning and long-context modeling:

**Video-MME**Fu et al. (2024) is a video evaluation benchmark designed for general video understanding. It contains 900 videos (256 hours in total) across various categories and durations, annotated with 2,700 expert-curated multiple-choice QA pairs. The dataset is partitioned into short (<2 min), medium (4–15 min), and long (30–60 min) subsets, enabling a detailed analysis of temporal scalability.

**LongVideoBench**Wu et al. (2024b) focuses on long-context multimodal reasoning. It comprises 3,763 videos of up to 1 hour in length and 6,678 annotated questions across 17 categories. The benchmark emphasizes fine-grained temporal retrieval and localized event reasoning, making it ideal for evaluating absolute time comprehension.

**LVBench**Wang et al. (2024b) is one of the most challenging benchmarks for long video understanding, with an average video length of over 4,000 seconds. It provides 1,549 QA pairs including multiple tasks such as entity tracking, temporal grounding, and causal reasoning, offering a comprehensive testbed for temporal-aware video modeling.

**Implementation Details** We adopt Qwen2.5-VL (7B and 72B)Bai et al. (2025) as our baseline model. For fair comparison and reproducibility, we utilize the publicly released checkpoints and re-evaluated all benchmarks following their official technical report. Our DATE also follows the same settings. In the evaluation, the baseline adopts a uniform sampling rate of 4 FPS, with the resolution set to 448 (longest side) and a maximum of 256 input frames across all benchmarks. All the experiments are conducted with Nvidia A100-80G GPUs. For our proposed TASS, deepseek-v3Liu et al. (2024) is used for caption generation. Then, the frames are extracted with 1 FPS for all videos to calculate the visual-textual similarity score with the generated caption. Visual-textual similarity is computed using the CLIP ViT-B/32Radford et al. (2021) model to enable the semantic-aware frame filtering. In the

Table 1: Performance comparison on long video benchmark with SOTAs, including Video-MME (w/o subtitles), LongVideoBench, and LVBench. For fairly comparison, we re-test the model based on the technical report disclosed by QwenVL team, with all video inputs preprocessed based on 4FPS and 448 resolution. (♠: official reported results. ♣: we re-test results). In the test, we found that the metric reported by QwenVL team on LongVideoBench were tested at 224 resolution. More experiments on different model could be found in AppendixC.

| Models | Size | Frames | Video-MME (w/o sub) | | LongVideoB | LVBench |
|---|---|---|---|---|---|---|
| | | | Long (30-60min) | Overall (0-60m) | val (8s-3600s) | val (avg.>4000s) |
| *Closed Video MLLMs* | | | | | | |
| GLM-4V-Plus | - | 256 | - | 70.8 | - | 58.7 |
| GPT-4o | - | 384 | 65.3 | 71.9 | 66.7 | 27 |
| Gemini-1.5-Pro | - | 1/0.5fps | 67.4 | 75 | 64 | 33.1 |
| *Open-source Video MLLMs>70B* | | | | | | |
| LLaVA-OneVision-72B | 72B | 32 | - | 66.2 | 61.3 | - |
| LLaVA-Video | 72B | 64 | 61.5 | 70.6 | 61.9 | - |
| Qwen2-VL | 72B | 768 | 62.2 | 71.2 | 60.4 | 41.3 |
| InternVL2.5-78B | 78B | 16-64 | - | 72.1 | 63.6 | - |
| InternVL3-78B | 78B | 16-64 | - | 72.7 | 65.7 | - |
| Qwen2.5-VL-72B♠ | 72B | 768 | - | 73.3 | 60.7 | 47.3 |
| Qwen2.5-VL-72B♣ | 72B | 256 | 63.4 | 72.7 | 66.9 | 48.8 |
| **DATE-72B(Ours)** | 72B | 256 | **65.3** | **73.3** | **68.1** | **52.1** |
| *Small Video MLLMs* | | | | | | |
| VITA-1.5 | 7B | 16 | 47.1 | 56.1 | - | - |
| LLaVA-Video | 7B | 64 | - | 63.3 | 58.2 | - |
| NVILA | 8B | 256 | 54.8 | 64.2 | 57.7 | - |
| ByteVideoLLM | 14B | 256 | 56.4 | 64.6 | - | - |
| VideoLLaMA3 | 7B | 180 | - | 66.2 | 59.8 | 45.3 |
| InternVL3-8B | 8B | 16-64 | - | 66.3 | 58.8 | - |
| Qwen2.5-VL-7B♠ | 7B | 256 | - | 65.1 | $56.0_{224dpi}$ | 45.3 |
| Qwen2.5-VL-7B♣ | 7B | 256 | 55.4 | 65.8 | $61.8_{448dpi}$ | 43.7 |
| **DATE-7B(Ours)** | 7B | 256 | **57.3** | **67.3** | **63.3** | **47.4** |

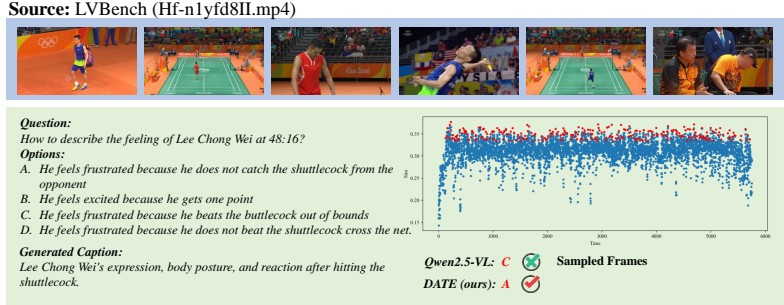

Figure 4: A real demo compared DATE-7B with Qwen2.5-VL-7B. The caption is generated with our method and calculate similiarity scores with frames. The **red** points are sampled frames with TASS. More could be found in **Appendix**.

TASS (Temporal-Aware Similarity Sampling) module, we set the selection ratio coefficient $\alpha = 4$, and initialize the temporal interval constraint $\delta_0$ to 20 seconds.

## 4.2 MAIN RESULTS

**Comparison with the State-of-the-Art** We compare our proposed method, DATE, with a variety of state-of-the-art closed-source and open-source video MLLMs on multiple long-video benchmarks, as summarized in Table 4. Compared to other small-scale video MLLMs, DATE achieves consistent improvements across all benchmarks, outperforming the prior best model (Qwen2.5-VL) by +1.5% on Video-MME (Overall), +1.5% on LongVideoBench (val), and +2.1% on LVBench (An extremely long video benchmark). Moreover, our method (256 frames) even outperforms the Qwen2.5-VL-72B

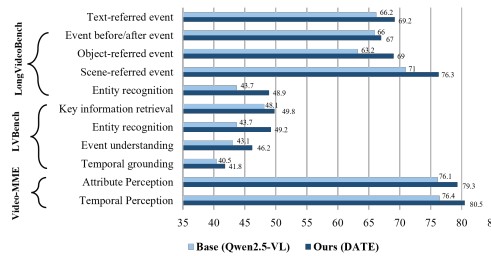

Figure 5: Comparison of performance related to event-aware tasks in the three benchmarks: Video-MME, LongVideoBench, and LVBench.

Table 2: Ablation study on two components of DATE-7B on three long video benchmarks: Video-MME, LongVideoBench, and LVBench.

| TIM | TASS | V-MME | LongVideoB | LVB |
|:---:|:---:|:---:|:---:|:---:|
| ✗ | ✗ | 65.8 | 61.8 | 43.7 |
| ✓ | ✗ | 66.5 | 61.9 | 44.9 |
| ✗ | ✓ | 66.6 | 62.8 | 46.7 |
| ✓ | ✓ | **67.3** | **63.3** | **47.4** |

Table 3: Comparisons with latest methods on LVBench. The baseline is the Qwen2.5-VL-7B model with uniform sampling and their MRoPE. **Sampling Strategy:** we compared TASS with AKS (most latest method), and list the computation time for both methods under the same CPU. **Time Embedding:** We compared our method TIM with timestamps given in prompt.

| Frames | Base | SamplingStrategy | | | | Time Embedding | |
|:---:|:---:|:---:|:---:|:---:|:---:|:---:|:---:|
| | | TASS(Ours) | | AKSTang et al. (2025) | | TIM(Ours) | Prompt |
| 256 | 43.7 | **46.7** | 21.2s | 45.8 | 21.1s | **44.9** | 42.5 |
| 128 | 40.7 | **45.8** | 6.4s | 44.6 | 19.2s | **40.2** | 39.4 |
| 64 | 38.8 | 42.6 | 2.7s | **43.3** | 16.4s | **37.1** | 36.9 |
| 32 | 36.8 | **40.9** | 1.7s | 39.6 | 13.9s | **37.3** | 35.8 |
| 16 | 33.9 | **39.8** | 1.2s | 33.8 | 11.7s | **35.7** | 33.1 |

(768 frames) model on LongVideoBench and LVBench. These gains demonstrate DATE's superior temporal modeling capability, especially in handling extremely long videos. It shows our methods effectively injects temporal cues and helps the model focus on semantically important moments, enabling more robust long-range reasoning.

**Comparison with Event-aware tasks.** To better understand the advantage of DATE in modeling temporal and event-centric information, we provide a detailed comparison across fine-grained sub-tasks in Video-MME, LVBench, and LongVideoBench, as shown in Figure 5.

### 4.3 PRECISE EVENT LOCALIZATION CAPABILITIES

Our DATE shows significant advantages in accurate event localization. As shown in the Fig.1, DATE can accurately identify the specific time points of events even when only 12 frames are used, and even accurately samples the critical time with only one frame as shown by the sampling order labeled in the sampling graph. However, the baseline model still shows significant deviations at 256 frames. This validates the effectiveness and robustness of our proposed temporal modeling and semantic-driven sampling strategy for long video understanding. Fig.4 also shows some cases in benchmarks, more examples can be found in the **Appendix**.

### 4.4 ABLATION STUDIES

We conduct comprehensive ablation studies to evaluate the two core components in DATE: Timestamp Injection Mechanism (TIM) and Temporal-Aware Similarity Sampling (TASS) on Video-MME, LongVideoBench, and LVBench, which are reported in Table 2.

To further analyze the effectiveness and efficiency of our sampling method, we compare TASS with Adaptive Keyframe Selection (AKS)Tang et al. (2025), a recent method proposed at CVPR'25, under large range of frame rates (**from 16 to 256**). As shown in Table3, TASS consistently outperforms AKS across nearly all frame settings, especially at lower frame counts (e.g., +6.0% at 16 frames),

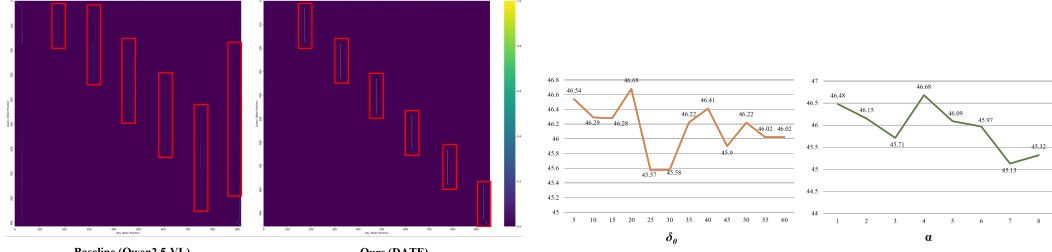

Figure 6: Attention maps of Qwen2.5-VL and our TIM with 6 times token. Rectangles label the attention area of each frame's vision tokens. TIM binds times to the corresponding frame and lead to a scope constraint on attentions.

Figure 7: Hyper-parameters analysis of TASS. $\delta_0$ is the initial minimum time interval for sampling, and $\alpha$ controls the candidate sampling frames.

while achieving comparable or even faster sampling times on the same CPU. These results highlight the efficiency and effectiveness of our sampling design.

Moreover, TIM consistently outperforms the simple "timestamp-in-prompt" method, demonstrating that directly embedding temporal cues into the token space is a more effective way to inject temporal awareness into MLLMs than relying on implicit prompt descriptions.

### 4.5 TIM ATTENTION ANALYSIS

To investigate the impact of temporal information on video understanding, we visualize attention maps of the baseline and our TIM. This experiment is conducted on the demo from Fig.1, using 12 input frames. Since Qwen2.5-vl merges every 2 frames, a total of 6 timestamp tokens are embedded.

As shown in Fig.6 (left), the baseline exhibits a relatively diffuse attention pattern, indicating that the model relies mainly on content-based similarity across the sequence. In contrast, the attention map of DATE (Fig.6, right) reveals a distinct pattern. Notably, video tokens corresponding to the timestamp receive significantly higher attention, suggesting that timestamp tokens act as temporal anchors. They enable the model to associate specific moments with the broader video content.

Furthermore, the explicit temporal cues introduced by timestamp tokens appear to improve the ability to localize frame information. By offering a temporal reference frame for aggregating content across the sequence, the model enhances its contextual understanding of individual video segments.

### 4.6 HYPER-PARAMETERS ANALYSIS

As shown in Fig.7 $\alpha$ controls the number of candidate frames, acting as an effective filtering mechanism to remove distracting information, it achieves the best performance at 4; $\delta_0$ constrains the initial temporal range of sampling, demonstrating the stability of the algorithm, which samples well no matter how it is initialized, ensuring continuity between frames and enhancing coverage of key events. Experimental results demonstrate that with appropriate configurations, TASS achieves a good balance between efficiency and temporal awareness.

### 5 CONCLUSION

In this work, we propose DATE, designed to enhance absolute time understanding and event localization in long videos for Multimodal Large Language Models (MLLMs). By timestamp tokens injection mechanism (TIM) and a semantic-driven key event sampling strategy (TASS), our method constructs an explicit and continuous temporal coordinate model with a Plug-and-Play way. Extensive experiments on multiple long-video benchmarks demonstrate that DATE significantly improves the model's ability to identify and align over temporally grounded events. Our findings highlight the importance of precise time modeling and open new direction to enhance time awareness for MLLMs.

## 6 REPRODUCIBILITY STATEMENT

Our method is a plug-and-play method, so that everyone can reproduce the same results as shown in the paper.

## 7 ETHICS STATEMENT

We do not encounter any ethical concerns, as our work is conducted entirely on publicly available models and benchmarks:

- **Benchmark:** Video-MME (Allows to used for academic research)
- **Benchmark:** LongVideoBench (CC-BY-NC-SA 4.0 license)
- **Benchmark:** LongVideoBench (CC-BY-NC-SA 4.0 license)
- **Model:** Qwen2.5-VL (Apache-2.0 license)
- **Compliance:** No private or proprietary assets were used. All usages comply with academic research standards and ethical guidelines.

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

## A  THE USE OF LARGE LANGUAGE MODELS(LLMs)

We used LLMs to assist in language polishing and grammar checking during the submission of the manuscript.

## B  ISSUES OF MRoPE AND PROOF

TIM primarily solves the MRoPE issue of Qwen2.5vl in a training-free way, meanwhile enabling the understanding of absolute time.

1. For intervals shorter than 1 second, it directly rounds the value (e.g., frames corresponding to 0.5s and 0.6s are encoded sequentially as $a, a+1$), which completely loses the concept of absolute time.

2. Its absolute time encoding shows almost no capability for absolute time perception. Additionally, for long videos, it wastes a large number of position IDs and causes relative positional ambiguity.

### B.1  RELATIVE POSITIONAL AMBIGUITY OF MRoPE IN QWEN-2.5VL

In long video understanding, sparse sampling is commonly used to reduce computation, such as sampling one frame per second. In this case, the position indices are often incremented uniformly (e.g., at seconds $0, 20, 40, \ldots$). Therefore, for Qwen2.5VL's MRoPE, **high-frequency dimensions** in RoPE suffer from **rotational aliasing**, which leads to **relative positional ambiguity**.

RoPE encodes each position $k$ by rotating its feature vector by an angle:

$$\phi_i(k) = k \cdot \theta_i, \quad \text{where} \quad \theta_i = 10000^{-2i/d}$$

For any two positions $k_1, k_2$, the relative positional difference is represented by the angle difference:

$$\Delta\phi_i = (k_2 - k_1) \cdot \theta_i = \Delta k \cdot \theta_i$$

For a large position difference $\Delta k$, the frequency parameter $\theta_i$ decreases rapidly with increasing dimension $i$, which causes high-frequency dimensions to be more sensitive to position differences.

When the angle difference $\Delta\phi_i$ exceeds $2\pi$, i.e., when the rotation completes a full cycle, RoPE maps the positions $k_1$ and $k_2$ to the same phase, leading to a loss of relative positional information.

Let $d = 256$ and the position interval be $\Delta k = 20$s. We compare two representative dimensions:

**Low-Frequency Dimension** ($i = 127$)**:**

$$\theta_{127} = 10000^{-\frac{2 \cdot 127}{256}} \approx 10^{-1.98} \approx 0.0105$$

$$\Delta\phi_{127} \approx 20 \cdot 0.0105 = 0.21 \text{ rad}$$

Since the period of low frequencies is long, it can still be distinguished quite well.

**High-Frequency Dimension ($i = 0$):**

$$\theta_0 = 10000^{-\frac{2 \cdot 0}{256}} = 1$$
$$\Delta\phi_0 = 20 \cdot 1 = 20 \text{ rad}$$

In this case, the angle difference $\Delta\phi_0 = 20$ rad corresponds to approximately $\frac{20}{2\pi} \approx 3.18$ full rotations. This may cause the relative position between two consecutive frames to become blurred.

## B.2 CRITICAL VALUE FOR MAINTAINING RELATIVE POSITIONAL RELATIONSHIP

In RoPE, the position encoding is given by rotating the feature vector by an angle $\phi_i(k)$ for position $k$. The frequency parameter $\theta_i$ for dimension $i$ is defined as:

$$\theta_i = 10000^{-2i/d}$$

where $d$ is the total number of dimensions, and $i$ is the index of the current dimension.

For the highest-frequency dimension ($i = 0$), the frequency is maximum:

$$\theta_0 = 10000^0 = 1$$

Thus, for $i = 0$, the rotation angle for a given position difference $\Delta k$ is:

$$\Delta\phi_0 = \Delta k \cdot \theta_0 = \Delta k$$

The rotation angle $\Delta\phi_0$ must remain within one period, i.e., within $2\pi$ radians. Therefore, we require:

$$\Delta\phi_0 < 2\pi$$

Substituting $\Delta\phi_0 = \Delta k$, we get:

$$\Delta k < 2\pi \approx 6.2832$$

For long videos, according to the Qwen2.5-vl method, the sampling interval can easily exceed this limit.

## C EXPERIMENTS ON OTHER MODELS

Table 4: Performance comparison on long video benchmark base on different models, including Video-MME (w/o subtitles), LongVideoBench, and LVBench. For fairly comparison, we re-test the model, with all video inputs preprocessed based on 4FPS and 448 resolution, and chose the supported 64-frame limit for sampling (both LLAVA-onevision and InternVL3 are up to 64 frames). (♠: official reported results. ♣: we re-test results). For InternVL3, we use their official inference codes, but it has a significant gap compared with their reported results

| Models | Size | Frames | Video-MME (w/o sub) | | LongVideoB | LVBench |
| | | | Long | Overall | val | val |
| | | | (30-60min) | (0-60m) | (8s-3600s) | (avg.>4000s) |
|---|---|---|---|---|---|---|
| LLAVA-onevision-7B♠ | 7B | 64 | - | 58.2 | 56.3 | - |
| LLAVA-onevision-7B♣ | 7B | 64 | 47.88 | 57.90 | 40.54 | 57.14 |
| w/ DATE | 7B | 64 | 48.11 | 58.62 | 45.12 | 58.56 |
| InternVL3-8B ♠ | 8B | 64 | - | 66.3 | 58.8 | - |
| InternVL3-8B ♣ | 8B | 64 | 50.55 | 61.81 | 54.90 | 43.45 |
| w/ DATE | 8B | 64 | 53.00 | 62.44 | 59.31 | 46.03 |

## D LIMITATIONS

Although DATE is an effective approach for enhancing absolute temporal understanding, it still encounters efficiency challenges when dealing with extremely long videos. The reliance on frame-level similarity computation and greedy selection under temporal constraints leads to an inference time that grows approximately linearly with video length. This may result in noticeable latency for hour-long videos—though such delays primarily occur during the initial pass, and subsequent interactions can leverage cached results for near-instant sampling. While reducing the sampling FPS can improve speed, it inevitably compromises precision. Future work may explore more scalable sampling strategies or hierarchical indexing mechanisms to improve runtime efficiency without sacrificing the model's ability to locate temporally critical events.

# E  TASS DEMO

This is the detail sampling visualization of Fig.2, with 16 sampled red points and sampling orders labeled.

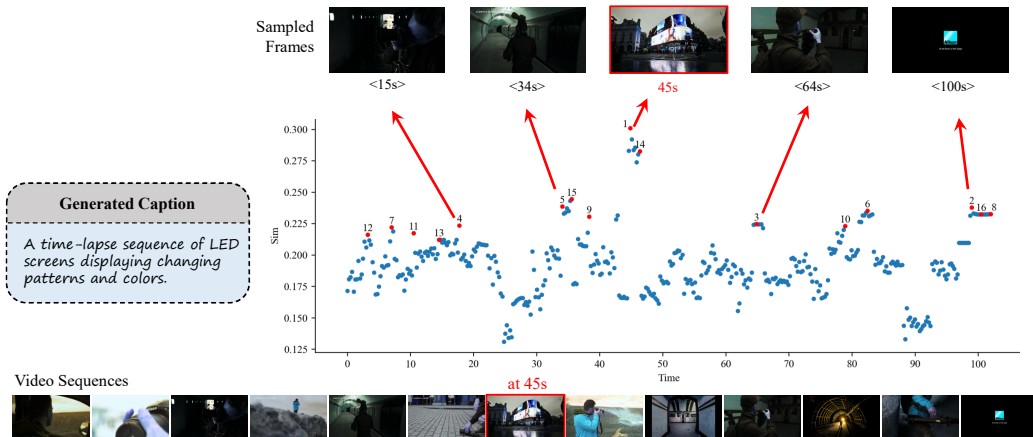

Figure 8: Sampling visualization.

# F QUALITATIVE RESULTS AND ANALYSIS

We present qualitative results to show the abilities of DATE-7B compared with Qwen2.5-VL-7B across various video understanding benchmarks. Fig.9,10,11,12,13,14 shows qualitative results on Video-MME, LVBench, and LongVideoBench.

**Source:** Video-MME (cy40DIzOUow.mp4)

**Question:**
*What is the third baked food in the video?*
**Options:**
A. Scallop.
B. Kobe beef.
C. Bacon.
D. Salmon.

**Qwen2.5-VL:** *B* ✗
**DATE (ours):** *C* ✓

**Generated Caption:**
*A plated dish with a baked item, positioned third in a sequence of foods.*

**Source:** Video-MME (cy40DIzOUow.mp4)

**Question:**
*In which period does the home team overtake the guest team?*
**Options:**
A. 12:56 - 8:13.
B. 5:58 - 2:57.
C. 8:13 - 5:58.
D. 13:10 - 10:37.

**Qwen2.5-VL:** *C* ✗
**DATE (ours):** *B* ✓

**Generated Caption:**
*The home team's score surpasses the guest team's score during a segment of the game.*

**Source:** Video-MME (tXb_zrHp4H8.mp4)

**Question:**
*In which part of the video is the woman in the blue top interviewed?*
**Options:**
A. Cannot be determined.
B. The beginning of the video.
C. The middle part of the video.
D. The latter part of the video.

**Qwen2.5-VL:** *B* ✗
**DATE (ours):** *D* ✓

**Generated Caption:**
*A woman in a blue top is seated, speaking to an interviewer in a studio setting.*

Figure 9: Qualitative Results on Video-MME compared with Qwen2.5-VL-7B (1).

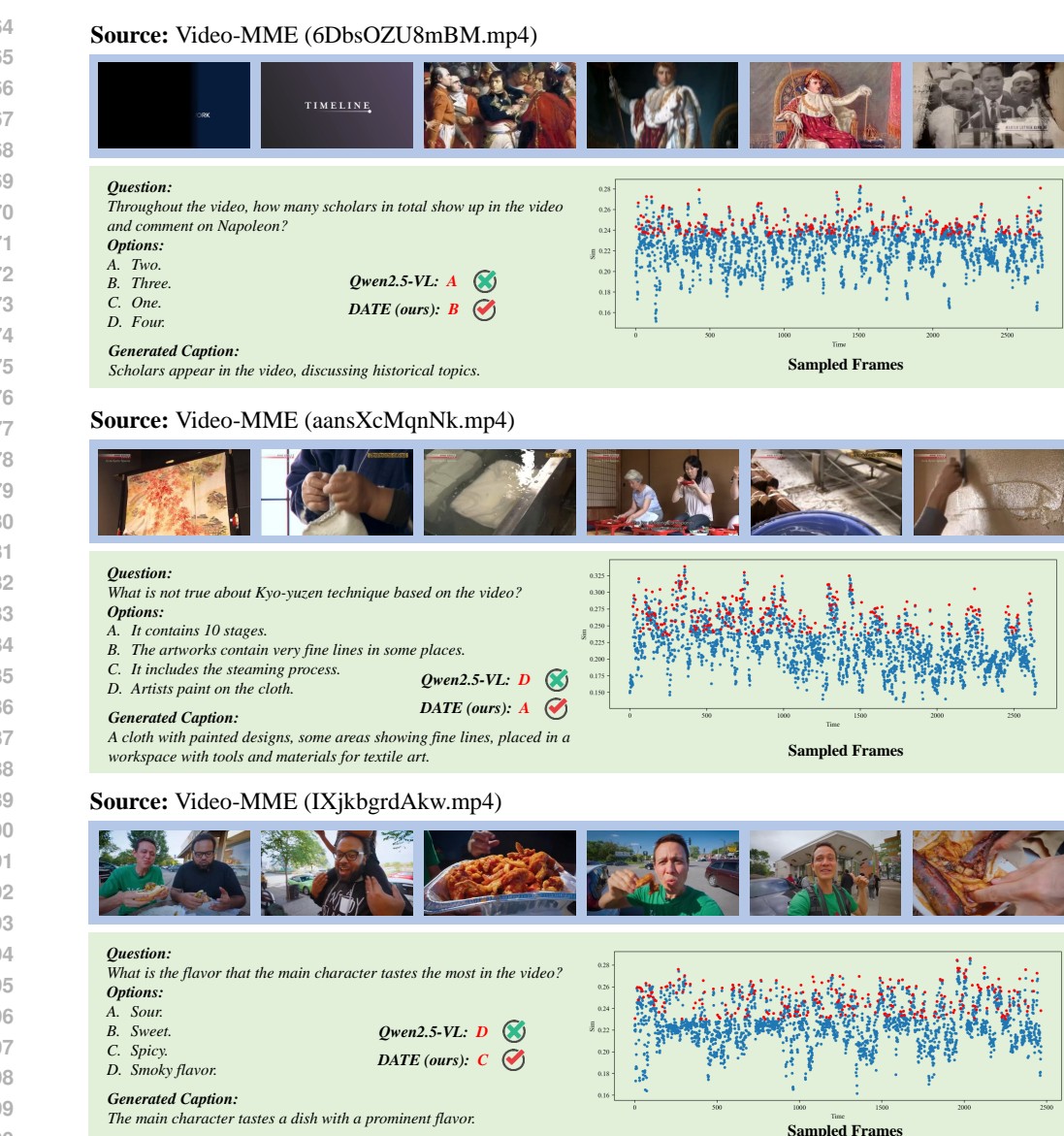

Figure 10: Qualitative Results on Video-MME compared with Qwen2.5-VL-7B (2).

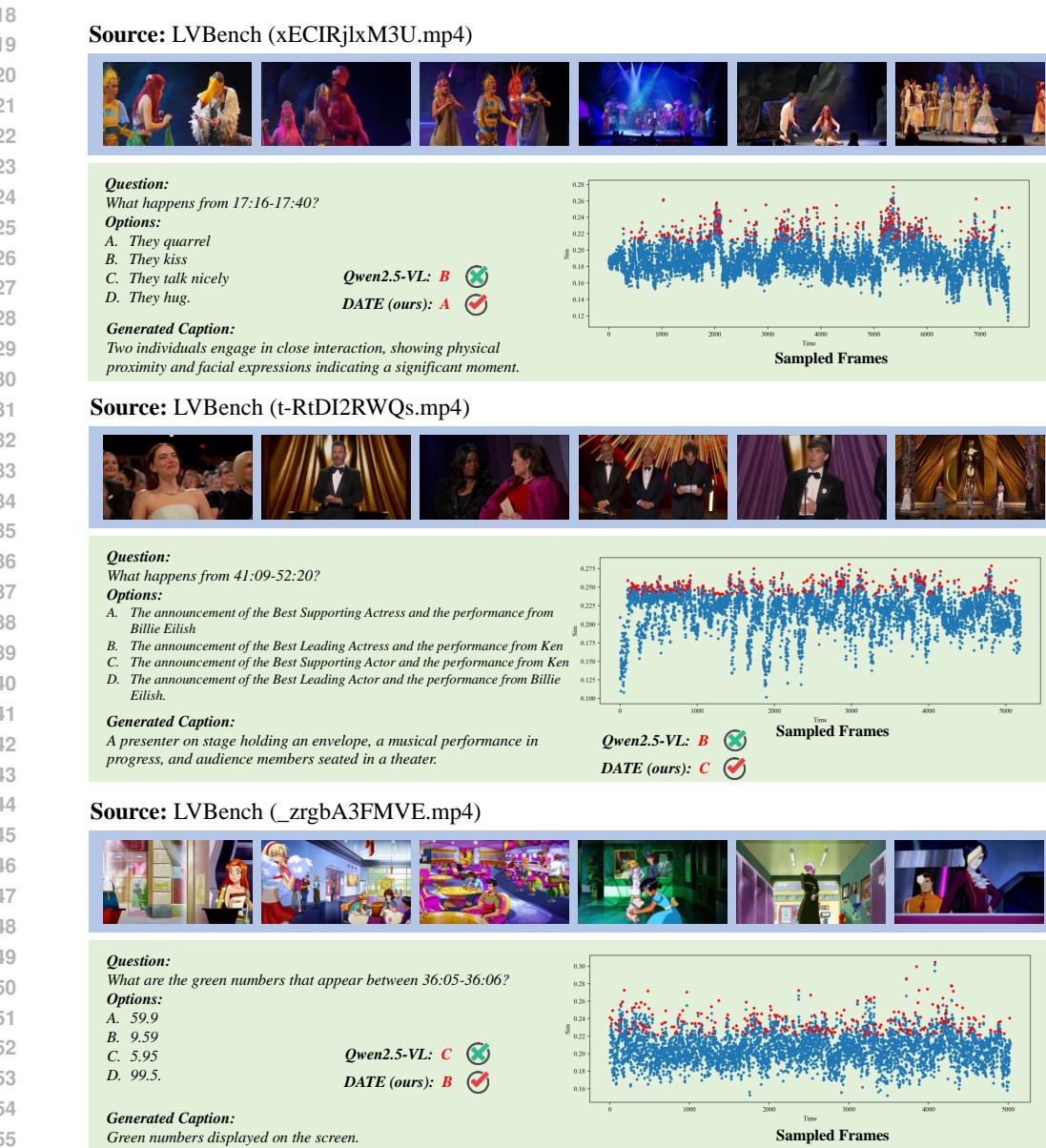

Figure 11: Qualitative Results on LVBench compared with Qwen2.5-VL-7B (1).

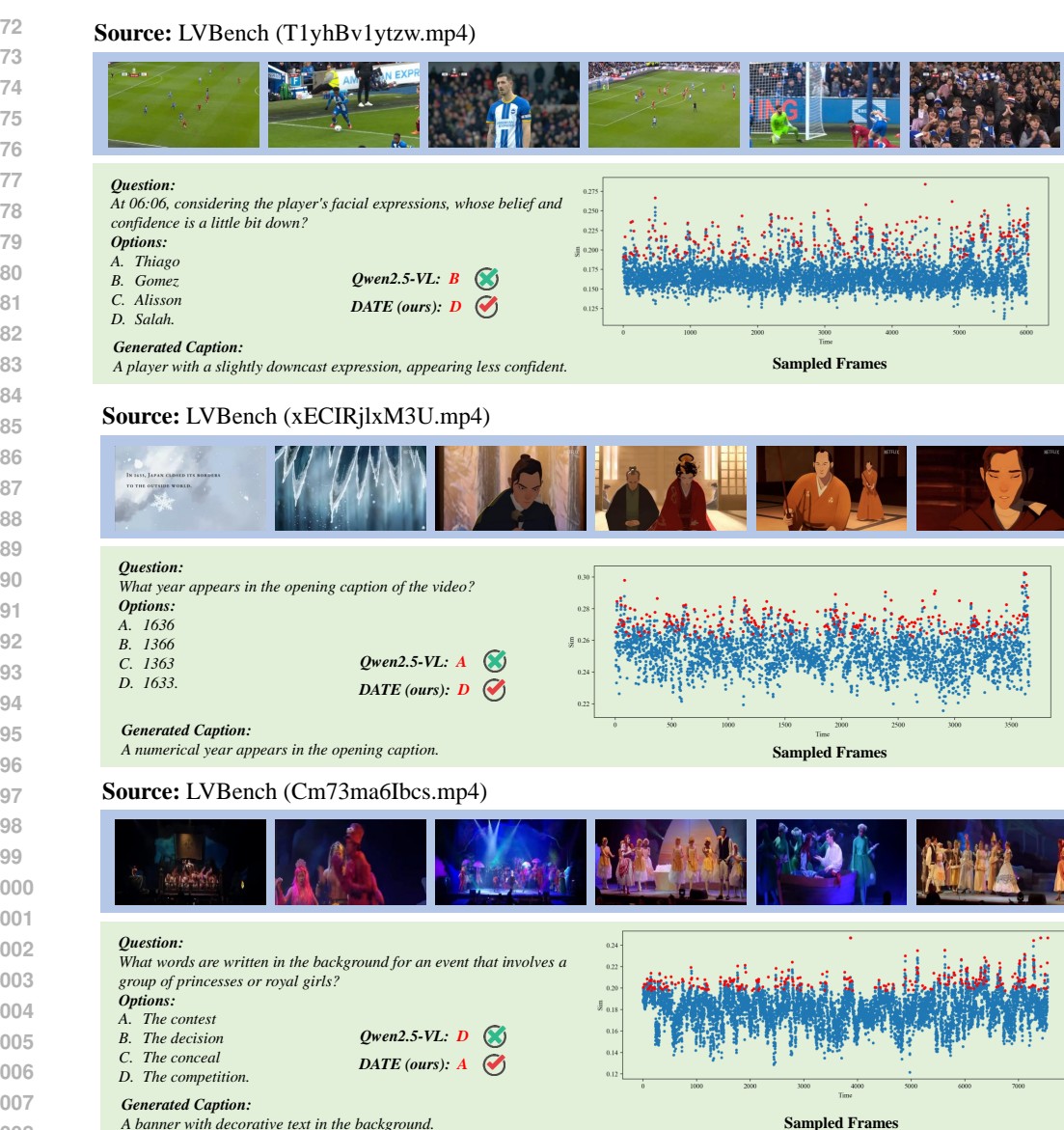

Figure 12: Qualitative Results on LVBench compared with Qwen2.5-VL-7B (2).

**Source:** LongVideoBench (vVRC-0VKPrg.mp4)

*Question:*
*The top of the screen shows a red search box, below the search box is left-aligned text, with bold black characters at the bottom. Some characters in the upper right center are on a blue background. In the lower right corner, there is a man in black wearing sunglasses explaining something using a speech bubble. What is this man doing?*
*Options:*
A. *Using the mouse pointer to add a green background to part of the text.*
B. *Using the mouse pointer to select part of the text on the screen.*
C. *Using the mouse pointer to add a red background to part of the text.*
D. *Using the mouse pointer to add a yellow background to part of the text.*
E. *Using the mouse pointer to add a white background to part of the text.*

*Generated Caption:*
*A man in black wearing sunglasses gestures with a speech bubble in the lower right corner.*

*Qwen2.5-VL:* **D** ❌
*DATE (ours):* **B** ✅

**Source:** LongVideoBench (eDso3zHFxL8.mp4)

*Question:*
*The individual on the screen is a man wearing a black hat and a white T-shirt. To his right, there is a photo that shows the back silhouette of a person looking at a sculpture. On the left side, there is also a yellow building. What is the man in the screen doing?*
*Options:*
A. *Raising both hands and facing away from a mirror while talking.*
B. *Raising both hands and looking up while talking.*
C. *Raising both hands and nodding while talking.*
D. *Raising both hands and looking down while talking.*
E. *Raising both hands and facing a mirror while talking.*

*Generated Caption:*
*A man in a black hat and white T-shirt raises both hands while talking.*

*Qwen2.5-VL:* **E** ❌
*DATE (ours):* **B** ✅

**Source:** LongVideoBench (d5JlCEDlHGE.mp4)

*Question:*
*Outside the room, there are two brown doors on a white wall. In front of the doors, there's a woman in gray clothes and a silver railing. On the wall, there's also a lamp emitting white light. What is this woman doing?*
*Options:*
A. *walking.*
B. *dancing.*
C. *talking.*
D. *running.*
E. *reading.*

*Generated Caption:*
*A woman in gray clothes stands in front of brown doors on a white wall, near a silver railing and a white lamp.*

*Qwen2.5-VL:* **C** ❌
*DATE (ours):* **A** ✅

Figure 13: Qualitative Results on LongVideoBench compared with Qwen2.5-VL-7B (1).

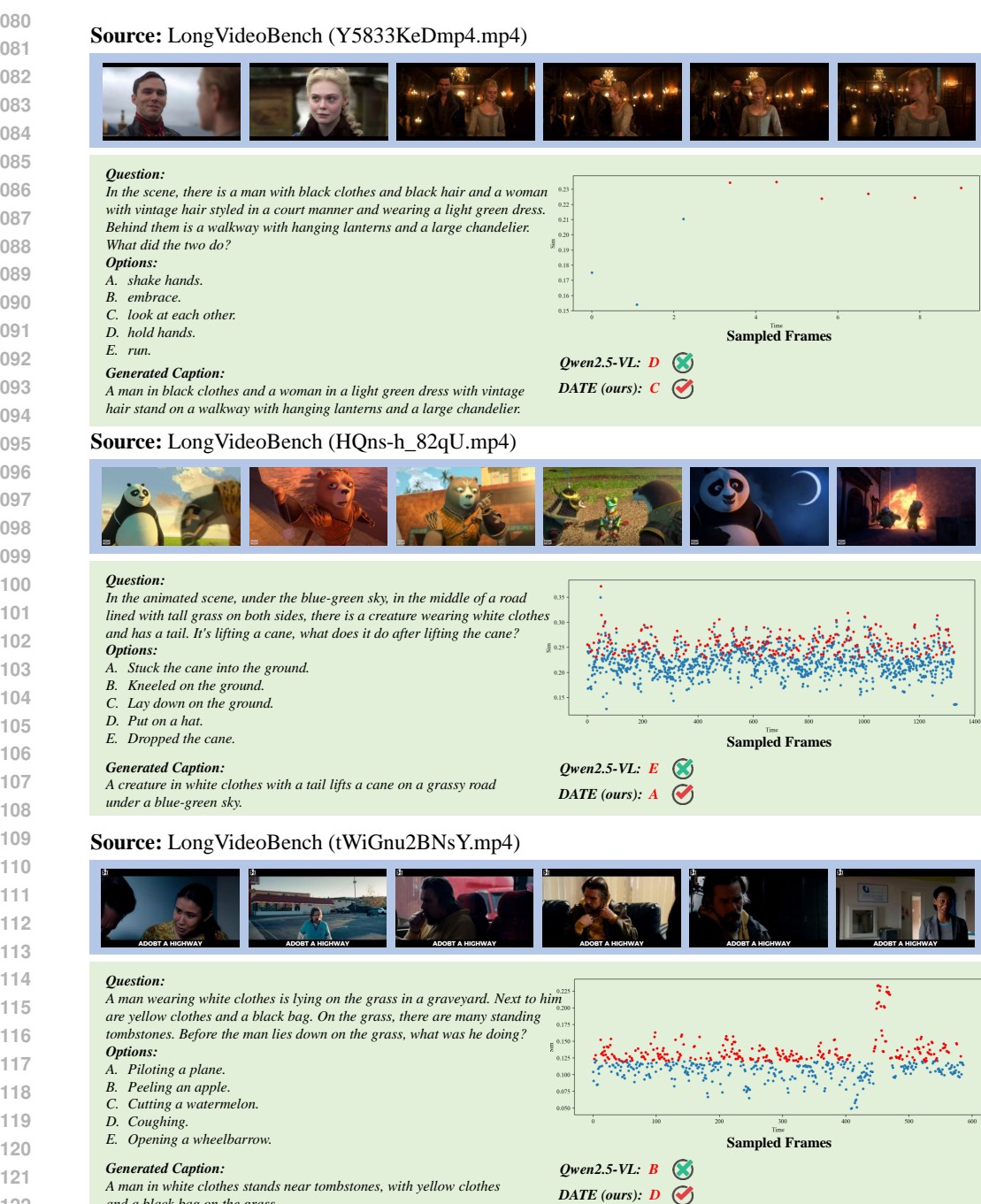

Figure 14: Qualitative Results on LongVideoBench compared with Qwen2.5-VL-7B (2).

## G  BAD CASES

While we obtained good boosts across the three benchmarks, we instead made errors compared to the baseline predictions in some cases, as shown in Fig.15. We believe this may be due to the fact that we introduced additional tokens that increased the processing difficulty of the model, bringing it close to the upper limit of its capacity, thus increasing illusions for certain scenario.

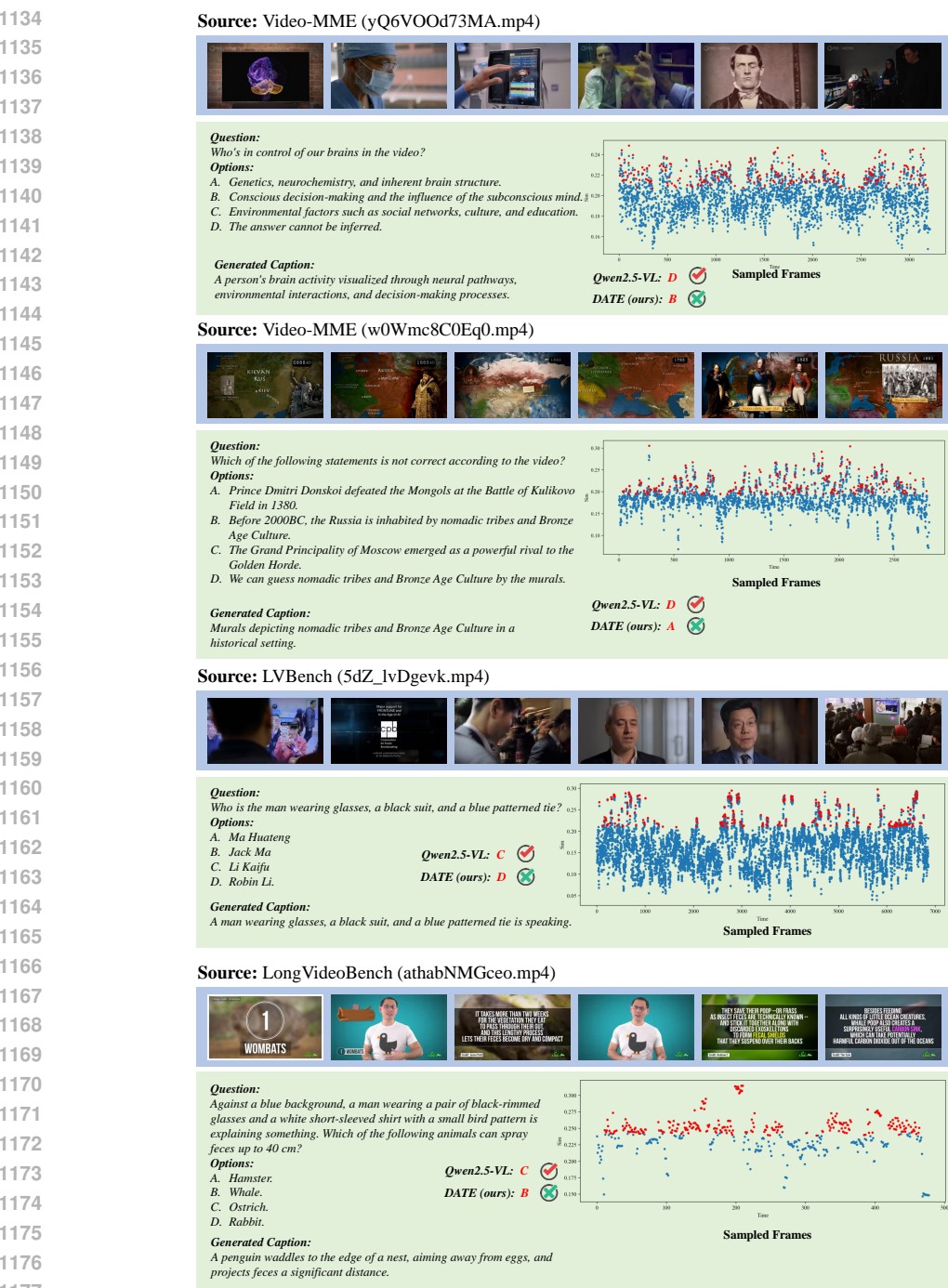

Figure 15: Bad cases compared with Qwen2.5-VL-7B.

## H  CAPTION GENERATION PROMPTS OF TASS

---

**Prompt**

You are an image description assistant. Assume you are currently watching a video, and I will give you a question related to the video.
Your task is to generate potential image caption based on the question, which is able to find the key image to answer the question.

Core requirements:
1. The output must be concise, objective, and visually observable facts.
2. Exclude subjective judgments, invisible information, and the specific content the question is asking.
3. Avoid using quantities; use implicit references instead.
4. The question options given are for reference, you can use their commonalities, but not only one of them.
5. Keep the output within 30 words.

Output format:
Directly output the visual description without any explanations or annotations.

Here is the question: {question}
Output Key Image Caption:

---

## I  DEMO WITH DIFFERENT TRAINING-FREE TIME EMBEDDINGS

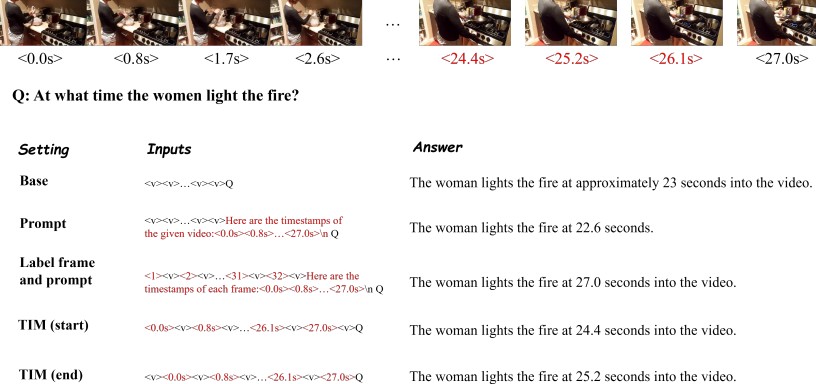

Figure 16: Compare TIM with MROPE (base), timestamps given in prompts, mark frames and given timestamps in prompt, TIM with time at the start of the vision tokens, and TIM with time at the end of the vision tokens. **It shows that both at the start/end of the vision segment is available for MLLM to understand a relatively absolute time.**

