# OpenReview forum: "DATE: Dynamic Absolute Time Enhancement for Long Video Understanding"
_ICLR.cc/2026/Conference — Submitted to ICLR 2026_

### Official Review · Reviewer_LxD8 · 2025-10-31

**Soundness:** 3
**Presentation:** 3
**Contribution:** 2
**Rating:** 4
**Confidence:** 5

**Summary:**

The paper proposes DATE, a training-free method to improve long-form video understanding of MLLMs through two light-weight modules: Timestamp Injection Mechanism (TIM) and Temporal-Aware Similarity Sampling (TASS). TIM interleaves each frame’s visual tokens with an explicit textual time token to build an absolute temporal reference and TASS reformulates frame sampling as caption-based clip retrieval and frame selection through greedy, similarity-first search. The paper is evaluated on 3 long video benchmarks and achieves superior performance.

**Strengths:**

1.	The proposed approach is training free.

2.	Good performance gain on the evaluated benchmarks.

**Weaknesses:**

1.	Method -

  a.	The paper is an engineering effort of repurposing existing methods without strong technical novelty.

  b.	TIM is essentially timestamp token concatenation, an already well explored idea [1-4] repurposed as a module.

  c.	TASS is essentially BOLT [5] + AKS [6] with renamed components and minor cosmetic implementation changes.

2.	Experiments -

  a.	While both TIM and TASS like approaches are already existing, there is no absolute performance comparison against those methods [1-6], even though they are almost identical. While one ablation compares against AKS, comparison with [1-4] is necessary given “time perception” is the central claim.

  b.	Despite claiming to solve “absolute time perception,” the method is not tested on any temporal grounding task benchmarks (e.g QVHighlights, Charades-STA, ActivityNetCaptions etc). Right now, it’s not clear whether TIM enables time understanding or just hints it textually. Even though the mentioned datasets are not truly long video, it still would have been a good starting point to show TIM’s effectiveness in time perception. Additionally, there is no proof that explicit timestamp tokens improve compositional or relational temporal reasoning.

  c.	Despite calling TASS “efficient,” the preprocessing of using DeepSeek and CLIP over every frame at 1fps is computationally expensive. How does the method scale with frames (e.g MovieChat-1k can grow as much as 10k+ frames). Additionally, the paper reports no runtime scalability, memory, FLOP etc to prove efficiency. Just showing CPU comparison to show efficiency is not enough.

  d.	While ablation isolates the performance contribution of TIM/TASS, there is no deeper exploration of these components. For example, What happens if timestamp tokens are randomized or inconsistent? How does timestamp density affect performance?

  e.	Using DeepSeek for caption rewriting introduces unfair comparison with other models that don’t use captioning models and this dependency is unaccounted for. How much does caption rewriting actually help beyond raw question embedding?

Minor Weakness

 1.	Extremely small fonts in figures are hampering readability.

References:

 [1] Ren et al. Timechat: A time-sensitive multimodal large language model for long video understanding. CVPR 2024

 [2] Chen et al. Timemarker: A versatile video-llm for long and short video understanding with superior temporal localization ability. arXiv 2024.

 [3] Guo et al. Vtg-llm: Integrating timestamp knowledge into video llms for enhanced video temporal grounding. AAAI 2025

[4] Huang et al. Lita: Language instructed temporal-localization assistant. ECCV 2024.

[5] Liu et al. BOLT: Boost Large Vision-Language Model Without Training for Long-form Video Understanding. CVPR 2025.

[6] Tang et al. Adaptive keyframe sampling for long video understanding. CVPR 2025.

**Questions:**

Weaknesses are mentioned in order of priority. Please refer to weakness for questions.

---

> ### Author Response · Authors · 2025-11-16
> **Response to Official Review by Reviewer LxD8 (1/2)**
>
> Thank you for pointing out our potential weaknesses. The detailed explanations are shown below.
>
> # Weakness:
>
> ## Method-a:
>
> **For TIM:**
>
> This is indeed a simple idea, and **it seems like it has definitely been discussed before, but it hasn't**. Existing methods don't embed time info in this way either. As some of literature you mentioned uses **additional training modules** or **adds watermarks** to introduce time information, which also increases the illusion of the model.
>
> **Motivation:**
>
> We want to design a plug-and-play method to train-free label the time of frames. However, related works were using **additional modules to encode time** and fusing it to vision embeddings with concatenation or cross-attention, or **labeling it by adding watermarks** to frames. We believe these are **inelegant** and increases the probability of the **model illusions** (even directly give to model could lead to wrong answers: 0.9>0.10, let alone to implicit fuse it).
>
> **Contribution:**
>
> - a. Most work treats the **video tokens as a whole**, rather than **counterintuitively inserting other tokens in between**. It's also generally believed that without fine-tuning, such token insertion will **cause the original hands to break down**. Moreover, compared to Qwen3VL (later than ours), which simply apply similar logic, our contribution goes beyond simply doing this; we **detailed analyze through attention map and VLM patterns why it works**, or **even work with training-free way**.
>
> - b. Due to the success of Qwen2.5VL, many people believe that their MROPE of absolute time can bring benefits to time understanding, but we **demonstrated its problems and provided a detailed explanation**. We **derive a relative token-awareness range of 6.28** from the perspective of ROPE (**Appendix B**), and thus analyze why Qwen2.5VL encounters relative position awareness problems when understanding very long videos.
>
> - c. Furthermore, we analyze in detail why implicit time representations, such as embedding absolute time through position encoding, are 1) **ineffective** in short video scenarios, and 2) cause **relative position perception problems** in long video scenarios due to ROPE. Based on this, we designed the TIM, combining it with the original MROPE. This mechanism can also be directly and seamlessly applied to various models.
>
> **For TASS:**
>
> 1) We are the **first to utilize captions to retrieve better keyframes** from our **image retrieval perspective**.
>
> 2) We designed for long video sampling. We balanced a)positive and negative samples, b)similarity, and c)temporal preservation to achieve a faster and better sampling method, especially suitable for long videos.
>
> 3) We achieved SOTA results for sampling methods.
>
> ## Method-b:
>
> TIM is a training-free method proposed based on MROPE, aiming to directly empower the original model by leveraging the modal alignment capabilities of MLLMs.
>
> However, references you mentioned:
>
> [1] It uses **self-attention to inject time** information. This requires additional training, and the actual effect of this implicit injection is poor (because explicit time injection models have an illusion: 0.9 < 0.10, let alone implicit ones).
>
> [2] It requires **timestamp token for training**, which requires a lot of training resources.
>
> [3] It gives **time information in prompt**, not time stamp embedding. **The last column of Tab.3 compared with it**.
>
> [4] The model outputs frame ids, and then replaces with timestamp. The entire timestamp is not involved in encoding, and it is not even entered into LLM. Moreover, **it cannot answer question like "what happened at 10:45".**
>
> ## Method-c:
>
> TASS is completely **different from [5][6] or anyelse works**, aside from the **similarity calculation step**, which is widely used in **all retrieval tasks**. So we could be confident to submit to a open review conference like ICLR.
>
> In fact, when we did this work we had no references at all. But we found them later, cited and compare them before submission.
>
> We conducted a **comprehensive comparison with AKS (Table 3)**, and duing the comparison, **AKS also use ours rewritten captions to sample**, highlighting the advantages of our method.

---

> > ### Author Response · Authors · 2025-11-16
> > **Response to Official Review by Reviewer LxD8 (2/2)**
> >
> > ## Exp-a:
> > Refer to **Method-b**. We are the **first time to implement training-free interleave embeddings**, something **no one has done before**[1][2][3][4] and **no training-free time embedding methods to be compared**. We demonstrate that this approach is superior to Qwen2.5vl's absolute-time MROPE.
> >
> > [1][2][4] are **train-required models**. Since no one done this training-free time embedding before, and we just need to prove our methods work and better than the baseline (Qwen2.5VL' MROPE).
> >
> > And since no training-free method like TIM, we compared a simple trianing-free way in Table3 with time given in prompts (like [3]). The time prompt is after vision tokens and combined with question input. Moreover, we have already tested on Qwen2.5VL(Table 1), LLaVA-ov(Appendix), and InternVL3(Appendix) to prove TIM works. **Qwen3VL also improved MROPE in subsequent using method like our TIM, proving our foresight and effectiveness.**
> >
> > Regarding sampling, we did not compare [5] because they are not open source even now (Nov. 2025), and their experimental results are not as good as [6].
> >
> > ## Exp-b:
> > We focused on understanding **hour-long videos**, and the selected benchmarks all contained videos longer than one hour. These benchmarks all **included temporal/event grounding tasks (Fig.5)**. Also, **LVBench** is a benchmark where **all QA (questions and answers) are related to hour-long video localization**, and we achieved a significant 4% improvement on it.
> >
> > Meanwhile:
> > - **QVHighlights:** we ask for the Permissions of their official Codalab Submission to evaluate but not recieved response yet.
> >
> > - **ActivityNetCaptions:** They directly **given the time and descriptions of key events into the input caption**. This allows the model to understand directly **without visual information**, making it impossible to prove the effectiveness of our method. This evaluation paradigm didn't align with our approach (just as we didn't test Videomme with subtitles). It **wasn't purely visual understanding**; textual information was also involved.
> >
> > Moreover, we further applied our method to two temporal grounding benchmarks and achieved good improvements. Specifically, we followed the official Qwen2.5vl settings, sampling 256 frames at 4fps as the input model. For the E.T. Bench, we used its official evaluation code; for Charades-STA, we used lmms-eval for evaluation, and the results are as follows:
> >
> > **Table I：Resuts on CharadesSTA and ETBench**
> > ||||
> > |-|-|-|
> > ||Charades-STA(mIOU)|ET Bench(Acc ref)|
> > ||||
> > |Qwen2.5VL|43.6|50.3|
> > |Ours|48.8|51.7|
> > ||||
> >
> > ## Exp-c:
> > **Most of the time is related to video decoding**. We show the specific time of different frame inputs of a video below. In addition, similarity sampling requires feature extraction based on a CLIP-like model ([5][6] is the same, so this part of the time is the same). The CPU time we compare is the time for sampling after obtaining the similarity across 1,549 QA on LVBench, and this sampling process only needs to be performed on the CPU. Compared with AKS, we are faster and have better results. (AKS also use the rewritten captions to sample).
> >
> > **Table II：The sampling time composition of a single video(in second)**
> > |||||||
> > |-|-|-|-|-|-|
> > |Total Frames|Video decode|CLIP encode|Similarity compute|Sampling(TASS)|
> > |||||||
> > |512|3.15|0.56|0.31|0.009|
> > |1024|5.19|0.57|0.57|0.01|
> > |2048|5.65|0.64|1.09|0.01|
> > |4096|9.78|0.85|2.13|0.011|
> > |||||||
> >
> > **FLOPs on caption rewritten:** One call on **DeepSeekV3 costs 0.24 TFLOPs/token**. For outputting a 30-word caption, the maximum cost is **only 7.2 TFLOPs, so only ~0.37s/call** on A100GPU, it is real-time. For data like MovieChat-1k, it is necessary to balance speed and accuracy.
> >
> > **FLOPs on CLIP(VIT-B/32):** For float data type, a single image is 14.8 GFLOPs, for N frames, it is 14.8xN GFLOPs, For the similarity computation, it is about 2 * N * N * D  (where D is 512 in this CLIP). Therefore, N frames total FLOPs is about 14.8N + 1024*N*N*D/10^9 GFLOPs. For 512 frames, it is about 7.58 TFLOPs.
> >
> > ## Exp-d:
> > We tested performance of randomly shuffling timestamps:
> >
> > **Table III：Random timestamp tokens**
> > ||||
> > |-|-|-|
> > ||Charades-STA(mIOU)|ET Bench(Acc ref)|
> > ||||
> > |Base|43.6|50.3|
> > |+TIM|48.8|51.7|
> > |+TIM(random order)|16.7|43.1|
> > ||||
> >
> > ## Exp-e:
> > **About fair:** We considered this issue, so in Table 3, comparisons with AKS are performed both using the rewritten captions, ensuring fairness. If not AKS would be worser.
> >
> > In addition, you raised an excellent question, and we conducted related experiments, including effects of rewriting using different language models. The results are as follows:
> >
> > **Table IV：TASS with different caption generators**
> > |||||||||
> > |-|-|-|-|-|-|-|-|
> > ||Uniform (base)|Original query|deepseek-v3(we used)|o4-mini|gemini2.5-pro-preview|doubao-seed-1.6|deepseek-R1|
> > |||||||||
> > |VideoMME|55.44|55.33|57.33|56.78|57.11|57.11|57.67|
> > |LongVideoB|61.78|61.25|62.75|61.26|62.90|64.10|63.20|
> > |LVBench|43.71|45.38|46.54|46.86|46.54|46.09|46.61|
> > |||||||||

---

> > > ### Comment · Reviewer_LxD8 · 2025-11-24
> > >
> > > I thank the authors for the detailed response. My primary concern was technical novelty, and after going through the rebuttal and other reviewers' comments, it still remains the same.
> > >
> > > - TIM is essentially an instance of attaching timestamp tokens to frame tokens and feeding them into a standard MLLM. Even if the implementation differs (e.g., insertion vs. cross-attention, training-free vs. trained modules), these change how the idea is executed, not what the idea fundamentally is. Being training-free makes it an engineering choice, not a new temporal modeling concept.
> > >
> > > - Doing something similar before a larger model release (Qwen3) does not make the idea conceptually original, nor validate TIM's design. The question is not whether TIM works; the question is whether the underlying idea is new. That remains unestablished.
> > >
> > > - Since no direct training-free baselines are provided, stronger comparisons are essential to understand what TIM actually contributes. The single baseline (time prompt after visual token) used is too weak and does not test other simple interleaving variants like different token placements, grouping, or simple step markers. Without such baselines, it is unclear whether TIM’s gains is indeed coming from its interleaving timestamp placement.
> > >
> > > - TASS shows similar concern. Reformulating sampling as caption-based retrieval yields only marginal improvements (~1%) with huge cost over using the original query, suggesting that the supposed main contribution of caption-driven retrieval is not particularly impactful.
> > >
> > > - While the limitations of prior methods are highlighted, it would have been more compelling to show where they fail, by how much, and why the proposed approach helps in those specific cases. Without targeted failure analysis (of standard MLLMs and not just Qwen), compute–performance tradeoffs, and clearer evidence of what is uniquely effective in TIM or TASS, it is difficult to view these components as technically novel.
> > >
> > > Overall, TIM and TASS both remain within previously explored conceptual spaces, and the rebuttal does not provide evidence that these components introduce new temporal modeling ideas. While the empirical improvements are appreciated, the technical novelty in this work is on the weaker side. So, I tend to retain my original rating.

---

> > > > ### Author Response · Authors · 2025-11-25
> > > >
> > > > Thank you for your detailed reply! Below is a brief explanation, discussion, and some of my own insights.
> > > >
> > > > TIM is primarily an implementation based on the 6.28 limit obtained in Appendix B. Only **by using MROPE can the temporal modeling of vision tokens and time tokens be controlled within 6** (since temporal encoding of vision tokens of the same frame are the same). The so-called 'engineering choice' refers to a simple yet effective implementation, and I personally prefer a simple yet effective method when reviewing papers. Furthermore, based on my knowledge in the field of video understanding, I am certain that methods like TIM will be the best temporal modeling under the current MLLM structure.
> > > >
> > > > The approximately 7.2 TFLOPs resulting from the one-call rewrite of TASS is not very high. If one-call values ​​are high, the ‘Agent' dies.
> > > >
> > > > In addition, I believe that this similarity sampling + practical embedding modeling approach is the most effective paradigm for achieving accurate and reliable long video localization. Methods like RL or Agent may works with short videos but not long videos, and I look forward to other concise and effective methods to assist in the understanding of hour-long videos.
> > > >
> > > > Thank you for your responsible and in-depth review. Even if it's rejected, I enjoyed the review process with you and other reviewers.

---

### Official Review · Reviewer_EXdF · 2025-11-01

**Soundness:** 3
**Presentation:** 3
**Contribution:** 2
**Rating:** 4
**Confidence:** 4

**Summary:**

This paper introduces DATE (Dynamic Absolute Time Enhancement), a plug-and-play framework to improve the capabilities of Multimodal Large Language Models (MLLMs) in understanding long videos, particularly for tasks that require precise temporal reasoning and event localization. The proposed method consists of two main components: 1) The Timestamp Injection Mechanism (TIM), which explicitly inserts textual timestamp tokens alongside video frame embeddings to create a direct and absolute temporal reference system for the model. 2) The Temporal-Aware Similarity Sampling (TASS) strategy, a two-stage, content-aware frame selection algorithm. The authors conduct extensive experiments on three challenging long-video benchmarks, demonstrating that DATE significantly outperforms strong baselines like Qwen2.5-VL.

**Strengths:**

1. The paper is well-written. The proposed method is well-illustrated and easy to follow.

2. **Training-free and model-agnostic:** A major advantage of DATE is its "plug-and-play" nature. It enhances existing, pre-trained MLLMs without any need for costly fine-tuning or architectural modifications.

3. **Thorough analysis:** The paper is supported by solid analytical work, including comprehensive ablation studies that isolate the contributions of TIM and TASS. The attention map visualizations provide qualitative validation for TIM's mechanism, and the comparative analysis against the AKS sampling method further substantiates the superiority of TASS.

**Weaknesses:**

1. **The novelty is limited**: The Timestamp Injection Mechanism (TIM) has been proposed by Seed1.5-VL [1], and the similar ideas of Temporal-Aware Similarity Sampling (TASS) have been widely explored by previous work like AKS, VideoAgent [2], etc.

[1] Seed1.5-VL Technical Report

[2] VideoAgent: Long-form Video Understanding with Large Language Model as Agent

2. **Inference latency for very long videos:** The TASS component requires an initial pass to extract frames (e.g., at 1 FPS) and compute similarity scores across the entire video. As acknowledged in the appendix, this introduces an inference overhead that scales linearly with the video's duration. For hour-long or multi-hour videos, this initial latency could be a significant practical limitation for on-demand applications.

3. **Dependency on external components:** The TASS method's performance is dependent on an external LLM (deepseek-v3) to generate the descriptive caption. The quality of this caption is critical for identifying the most relevant frames. The paper does not analyze the sensitivity of the system to the choice of this LLM or how it handles potentially noisy or inaccurate captions, which could mislead the sampling process.

**Questions:**

1. **Comparison with fine-tuning approaches:** Your training-free approach is a significant strength. However, for context, how do you see DATE's performance and methodology comparing to methods that explicitly fine-tune a model with temporal tokens and timestamped video data? What are the key trade-offs in terms of performance, data requirements, and temporal localization precision?

---

> ### Author Response · Authors · 2025-11-16
> **Response to Official Review by Reviewer EXdF (1/2)**
>
> Thank you very much for your careful reviews and valuable feedbacks. Here are the responses of your concerns.
>
> # Weakness:
>
> ## W1:
>
> **For TIM:***
>
> First, Seed1.5-VL is later than our work. We did this work within a month of Qwen2.5VL's release. Furthermore, **Qwen3VL also adopted a similar approach to TIM**, removing absolute time position encoding from MROPE and embedding interleave timestamps, demonstrating the **foresight of our approach**. Moreover, differ from Seed1.5-VL and Qwen3VL that simply do this for training without any analysis, we directly leveraged our insights into MLLMs to achieve this operation in a **training-free manner** and detailed analyze through attention map, RoPE and VLM patterns why it works.
> **Breaking vision tokens with Text seems like a simple operation, but counterintuitive.**
>
> **Motivation:**
>
> We wanted to design a plug-and-play method to train-free label the time of each frame. However, related works were using **additional modules to encode time** and fusing it to vision embeddings with concatenation or cross-attention, or **labeling it by adding watermarks** to frames. We believe these are **inelegant** and increases the probability of the **model illusions** (even directly give to model could lead to wrong answers: 0.9>0.10, let alone to implicit fuse it).
>
> **Contribution:**
>
> - a. Most work treats the **video tokens as a whole**, rather than **counterintuitively inserting other tokens in between**. It's also generally believed that without fine-tuning, such token insertion will **cause the original hands to break down**. Moreover, compared to Qwen3VL (later than ours), which simply apply similar logic, our contribution goes beyond simply doing this; we **detailed analyze through attention map and VLM patterns why it works**, or **even work with training-free way**.
>
> - b. Due to the success of Qwen2.5VL, many people believe that their MROPE encoding of absolute time can bring benefits to time understanding, but we **demonstrated its problems and provided a detailed explanation**. We **derive a relative token-awareness range of 6.28** from the perspective of ROPE (**Appendix B**), and thus analyze why Qwen2.5VL encounters relative position awareness problems when understanding very long videos.
>
> - c. Furthermore, we analyze in detail why implicit time representations, such as embedding absolute time through position encoding, are 1) **ineffective** in short video scenarios, and 2) cause **relative position perception problems** in long video scenarios due to ROPE. Based on this, we designed the TIM, combining it with the original MROPE. This mechanism can also be directly and seamlessly applied to various models.
>
> **For TASS:**
>
> - Our approach primarily combines **a) positive and negative samples, b) similarity, and c) temporal preservation** into a comprehensive design that integrates **retrieval, temporal analysis, and sampling**, achieving a balance to realize a faster and better sampling method, particularly suitable for long videos.
>
> - **The only similarity** to AKS or other 'similar' sampling methods is limited to the **calculation of similarity**, a step already **widely used in retrieval tasks**; everything else is different.
> Please believe that our work is fundamentally **different from related works**; otherwise, we wouldn't have chosen to submit to the **open-review conference ICLR**.
>
> **VideoAgent**, first, **doesn't involve sampling**. Second, it generates **captions for images** and uses LLM to determine their correctness. This approach requires **significant computational cost and time** to obtain the caption for each frame (we consider using VLM to generate captions for understanding, a kind of understanding-retrieval-reunderstanding approach, to be redundant). Our approach is the complete opposite: a Text2Image retrieval mode.
>
> ## W2:
>
> Yes, this latency is **mostly** due to **video decoding**, a **common issue** with video understanding **if you need more frames**. Higher precision sampling does require a longer latency; it's essentially a trade-off between performance and speed.
>
> **Table I：The sampling time composition of a single video(in second) on single A100-80G GPU**
> | | | | | | |
> |-|-|-|-|-|-|
> |Total Frames |Video decode (unsing ```decord```) | CLIP encode   | Similarity compute | Sampling (TASS) |
> | | | | | | |
> |512 |3.15 | 0.56|0.31|0.009 |
> | 1024 | 5.19 |0.57|0.57|0.01 |
> | 2048 | 5.65 |0.64|1.09|0.01|
> |4096 | 9.78 |0.85|2.13|0.011 |
> | | | | | | |
>
> Furthermore, in real system, this latency only exists in the **first round of question answering**; subsequent questions and answers can be sampled and understood in real-time based on cached features.

---

> > ### Author Response · Authors · 2025-11-16
> > **Response to Official Review by Reviewer EXdF (2/2)**
> >
> > ## W3:
> >
> > This is an issue that should be considered by **any work using LLM**, including **all agent papers** (such as **VideoAgent** mentioned above). While misinterpretation is unavoidable, a brief rewrite (30 words) like the one in this paper may **not introduce very significant error**.
> >
> > We also compared the impact of different LLMs, and the results are as follows. This can also reflect the capabilities of LLMs.
> >
> > **Table II：TASS results with different caption generators on three benchmarks**
> > | | | | | | | | |
> > |-|-|-|-|-|-|-|-|
> > | |Uniform (base)| Original query   | deepseek-v3 (used in this paper) | o4-mini | gemini-2.5-pro-preview | doubao-seed-1.6 | deepseek-R1 |
> > | | | | | | | | |
> > |Video-MME |55.44|55.33|57.33|56.78|57.11|57.11|57.67|
> > | LongVideobench | 61.78   | 61.25   | 62.75   | 61.26 | 62.90    | 64.10        | 63.20  |
> > | LVBench | 43.71   | 45.38  | 46.54   | 46.86 | 46.54   | 46.09  | 46.61   |
> > | | | | | | | | |
> >
> >
> > # Questions:
> >
> > ## Q1:
> >
> > **Overall:**
> >
> > - Fine-tune methods limitation: 1)**different models need to be retrained**; 2) for time-localization tasks, precise data annotation is required for supervised training, as errors in the dataset can affect model performance.
> >
> > - Ours: could be directly to any model because 1) MLLMs have the ability to align text and images, enabling direct understanding; 2) Ours can directly segment information from different frames and bind it to response visual tokens.
> >
> > **Detailed:**
> >
> > Currently, many timestamp-based fine-tuning methods **encode time through an encoder, inject time into visual tokens via cross-attention, or use watermarks on frames** (just like related works other reviewers point out). We believe it's better to directly display this timestamp to the model.
> >
> > Firstly, these indirect time embeddings, like Qwen2.5VL's absolute position encoding, involves feature stacking at the embedding level. This abstract approach prevents the model from grasping the information. Even if directly displayed to the model, LLM can still produce the illusion like 0.9 < 0.10, the indirect ones may introduce **more uncertainty and illusions**.
> >
> > Furthermore, some methods that **use labeled target time segments for fine-tuning have high requirements for labeled data**; labeling errors in some data can interfere with the model's judgment. Our tag-like approach **directly binds time and visual information**, allowing us to leverage the model's inherent capabilities for localization. Unlike models like Qwen2.5vl, Internvl3, and LLaVA-onevision we compared, which lack temporal information and with ours they directly have temporal location capabilities.
> >
> > **Moreover**, We believe that applying this method **followed by fine-tuning can yield better performance**. As Qwen3VL, they applied a **similar approach to ours** (removed the absolute time position encoding in MROPE and embedded interleave timestamps) and then trained the model, achieving great time-awareness performance. This also demonstrates the **effectiveness and foresight** of our approach.

---

### Official Review · Reviewer_f994 · 2025-11-05

**Soundness:** 2
**Presentation:** 3
**Contribution:** 2
**Rating:** 4
**Confidence:** 5

**Summary:**

This paper introduces explicit timestamp tokens into MLLMs to improve time-aware video understanding and event localization. Meanwhile, it employs a two-stage semantic-guided key frame selection. Experiments are conducted on general video understanding benchmarks (VideoMME, LongVideoBench, and LVBench) and their event-aware tasks.

**Strengths:**

- The paper is well written and easy to follow.
- The idea of injecting timestamp tokens into a video-language transformer is simple and intuitive.
- The proposed approach is plug-and-play — it can be applied to existing multimodal LLMs without dense temporal grounding annotations or extensive retraining, making it appealing for low-resource or training-free settings.

**Weaknesses:**

1. **Misalignment between problem definition and evaluation.**

The paper positions itself as addressing timestamp-aware video understanding and temporal/event grounding, yet the primary results are reported on general video understanding benchmarks (VideoMME, LongVideoBench, LVBench). Although some event-aware subsets are included, these benchmarks do not directly reflect timestamp reasoning or grounding capabilities.
To support the claimed contributions, evaluations on specialized temporal grounding datasets for MLLMs are necessary, such as:

- E.T. Bench (NeurIPS 2024) — open-ended event-level temporal grounding for video LLMs
- Charades-STA — standard video temporal localization benchmark

At present, the experiments do not sufficiently demonstrate that explicit timestamp modeling improves grounding over existing methods.

**2. Limited novelty; similar ideas have been explored.**

The main technical idea—inserting explicit timestamp tokens into the multimodal input stream—has been extensively explored in temporal grounding literature and even appears in open-sourced MLLMs (e.g., Qwen3-VL). As a result, the contribution feels incremental. Several recent works already introduce timestamp or temporal tokens into video-LLMs or temporal grounding pipelines:

Number It: Temporal Grounding Videos like Flipping Manga, CVPR 2025
GenS: Generative Frame Sampler for Long Video Understanding, ACL Findings 2025

Thus, timestamp token injection appears more like a commonly used trick rather than a novel methodological contribution.

**3. Missing discussion on reinforcement learning (RL) effects.**
Recent GRPO-based work (e.g., TimeR1: Qwen2.5-VL + GRPO) shows that reinforcement learning can significantly improve grounding capability even without explicit temporal embedding.
Therefore, it remains unclear whether the observed gains originate from:

- the timestamp embedding mechanism itself, or
- insufficient grounding training of the base model.

Ablations such as timestamp embedding vs. RL-enhanced grounding would clarify this.

**Questions:**

- In Table 3, does the reported runtime of TASS include the DeepSeek-V3 recaptioning step? If not, the latency comparison against AKS may not be fair.

- Why are general video understanding benchmarks used as primary evaluation instead of timestamp-aware grounding datasets such as ETBench or Charades-STA?

- If reinforcement learning (e.g., GRPO) is added to the base model, do timestamp embeddings still yield improvements?

- Does the method require additional training data or annotations, or is it completely training-free?

---

> ### Author Response · Authors · 2025-11-16
> **Response to Official Review by Reviewer f994 (1/2)**
>
> We are grateful for your detailed reviews and constructive suggestions. In the following, we respond to the concerns you mentioned.
>
> # Weakness:
>
> ## W1:
>
> We focused on understanding **hour-long** videos, and the selected benchmarks (Videomme, LVbench, LongVideoBench) all contained videos **longer than one hour**. These benchmarks also included **temporary grounding tasks**. For example, LVBench is a benchmark where **all QA is related to hour-long video temporary/event localization**, and we achieved significant improvements 4% on it.
>
> However, for example, Charades-STA is **only 32 seconds long**, and it could be detailed understood without sampling using model like Qwen2.5VL (which can input 768 frames), and therefore cannot demonstrate the method's advantage.
>
> Below, we further applied our method to these benchmarks and achieved good improvements. Specifically, we followed the official Qwen2.5vl settings, sampling 256 frames at 4fps as the input. For the E.T. Bench, we used its official evaluation code; for Charades-STA, we used ```lmms-eval``` for evaluation, and the results are as follows:
>
> **Table I：Resuts on Charades-STA and ET Bench**
> | | | |
> |-|-|-|
> | |Charades-STA (mIOU)| ET Bench (Acc ref) |
> | | | | | | | | |
> |Qwen2.5VL |43.6 |50.3|
> | Ours | 48.8   | 51.7   |
> | | | |
>
> We would also like to cite and include them in the paper.
>
> ## W2:
>
> **Our nature is different from theirs**; our perspectives and the depth of our discussions are also different.
>
> 1. **Number it** uses video frame numbers as **watermarks** on images. This requires **additional processing time**, destroys some information in the original image, and **impairs OCR capabilities**.
>
> 2. **GenS** uses **VLM to score frames and queries**, which is an **extremely wasteful approach** in terms of computational resources and time. For thousands of frames, considering the input limit and the illusion caused by context length, this method needs to **call VLM hundreds of times** to obtain the score for each frame. Undoubtedly, there is no shortage of practical solutions.
>
> In fact, we did this work within a month of Qwen2.5VL's release. During testing, we discovered their **absolute time awareness failure** and proved that the proposed absolute time alignment MROPE, while **lacking absolute time awareness**, also affected the model's perception of the **relative position of tokens** (**Appendix B**). In Appendix B, We derived a **relative position awareness range of 6.28** and **designed the TIM mechanism within this range**, combining it with the original MROPE. This mechanism can also be seamlessly applied to various models using MROPE.
>
> **Similar logic as TIM is used in Qwen3VL** (later than ours), where they also removed the absolute time position encoding in MROPE and embedded interleave timestamps, demonstrating our **forward-thinking**.
>
> Morover, many people generally treat the entire video as an indivisible whole, ignoring MLLM's feature of **visual and text alignment**. This is why, as you mentioned, **"Number it"** clumsily uses the video **frame number** as a **watermark** on the image.
>
> We, however, verified that MLLM can train-free perceive interleave information embedding and visualized it in Fig.6, demonstrating that this TIM also has the function of **information segmentation**, thereby more accurately locating keyframes.
>
> ## W3:
>
> First, we demonstrated that Qwen2.5vl's MROPE **lacks absolute temporal grounding capabilities and may even have negative effects**, making it **impossible** to complete absolute temporal localization tasks (no matter how it's trained, it's impossible unless accurate absolute temporal injection is provided).
>
> >Why Qwen2.5VL cannot understand absolute time?
> >1. For short videos with high fps, their position id **cannot be float**, making it **impossible** to represent time.
> >2. For long videos, due to the inherent characteristics of ROPE, it cannot represent relative positions well if position ids of two adjacent tokens beyond 6.28(proved in Appendix B), making it hard to perceive relative positions, which is especially noticeable in very long sequences.
>
> Our main focus is on a plug-and-play way. Furthermore, we believe that TIM with post training can **yield even greater improvements**. As the similar modification of TIM in Qwen3VL, and based on our knowledgement and subsequent training experiences, we believe Qwen3VL's video performance improvements are **almost due to this interleaved time embedding like ours**. The sucessful of Qwen3VL also proved the effectiveness of this idea with post training.

---

> > ### Author Response · Authors · 2025-11-16
> > **Response to Official Review by Reviewer f994 (2/2)**
> >
> > # Questions:
> >
> > ## Q1:
> >
> > We have considered this issue, so in Table 3, the **comparison with AKS also uses the rewritten caption (the same of us)** for sampling, ensuring fairness. Without the rewriting, their methods would be even worser.
> >
> > Furthermore, caption rewriting costs **DeepSeekV3 0.24 TFLOPs/token**. For outputting a **30-word caption**, the maximum **cost is only about 7.2 TFLOPs**, resulting in only ~0.37s/call on a A100 GPU. This is very fast.
> >
> > ## Q2:
> >
> > Please refer to **W1**
> >
> > ## Q3:
> >
> > Yes. As Qwen3VL's modification of MROPE which is similar to our TIM, demonstrated, this method is indeed effective.
> >
> > TIM is a plug-and-play method. We've already packaged it, and you can directly **use two lines of code to replace the original function of Qwen2.5vl** in the ```transformers``` package to apply it.
> >
> > ## Q4:
> >
> > **No annotations or training are required at all**.
> > We were also puzzled at that time, since MLLM already aligns visual and textual information very well, why didn't anyone do this in a training-free manner like our TIM?
> >
> > As our motivation suggests, we want to leverage the **inherent capabilities of MLLMs** to achieve absolute time awareness in low cost way. And this is differ than other related methods (including other reviewer mentioned) that require **additional parameter module to embed time** and need **costly training**.

---

> > > ### Comment · Reviewer_f994 · 2025-11-17
> > > **After-Rebuttal Recommendation**
> > >
> > > I appreciate the authors’ efforts during the rebuttal period. I now fully understand the advantages of DATE as a plug-in module and have no remaining concerns regarding its ability to improve temporal-aware tasks for Qwen2.5-VL.
> > >
> > > After carefully considering the authors’ responses and reading the comments from other reviewers, I have decided **not to change my original score**. My reasons are as follows.
> > >
> > > The primary reason for not increasing my score remains the same: I believe the central contributions of the paper—timestamp-token injection and the clip-based frame sampler—are closer to commonly used tricks rather than a novel methodological contribution. As another reviewer also noted, these designs resemble “engineering practice” more than new scientific insight. **In terms of technical novelty, I do not think the current method meets the bar for acceptance at ICLR**.
> > >
> > > From a research perspective, **what would be more meaningful is a deeper and more rigorous analysis supported by quantitative pioneer experiments of why current MLLMs such as Qwen2.5-VL struggle with absolute temporal understanding**, and ideally extending this analysis to other mainstream MLLMs. This is an important and valuable problem. After that, the paper could briefly introduce the two proposed plug-in modules and summarize their empirical benefits, which would be sufficient given the scale of the contributions.
> > >
> > > Overall, I believe the paper would benefit from a more substantial re-organization, especially in Section 3 (Methods), which could follow the logical structure of the Introduction more closely and clarify the scientific problem "Why" before presenting the plug-in designs.
> > >
> > > For these reasons, I am keeping my original score.

---

> ### Author Response · Authors · 2025-11-17
> **Thank for Advice and Statements**
>
> Thank you for these kindful suggestions!  They are helpful but may too late.
>
> As you pointed out, with major companies (Qwen, Seed) increasingly adopting method similar to TIM, and more sampling methods emerging, we understand that no matter how deeply we explain the depth of our thinking, in the current flood of papers, most people may see it as a re-organization of existing methods. **Especially since Qwen3vl has already similarly improved upon the problems in Qwen2.5vl,** no one may pay attention to our work anymore, and no one may investigate why the Qwen2.5vl MROPE has problems (we explained and proved), since **"Ohh, Qwen does that, so it is what should be."**
>
> We know that with more and more models using methods like TIM, our paper is being submerged in the tide, with no chance of acceptance after ICLR. We regret not having placed a "placeholder" on ArXiv as soon as possible, but we are willing to post this thought on ArXiv, waiting for someone to appreciate it (may be no one).
>
> Finally, we reiterate our advantages:
>
> 1. We are the first to propose a interleaved training-free time annotation method (TIM). (no one done like this before, even as a 'trick' since they think this could break vision information)
>
> 2. Our proposed TASS achieves SOTA results in sampling methods. And it is about sampling not clip similarity. If using clip similarity is resemble of AKS, than AKS (CVPR25) is resemble of "TEMPCLR: TEMPORAL ALIGNMENT REPRESENTATION WITH CONTRASTIVE LEARNING" (2022).
>
> 3. We **believe** that for temporal understanding of long videos, this workflow (DATE: TASS+TIM), which **first samples key frames and then inputs LLM with timestamp embedded between visual tokens** is the **optimal solution for understanding hour-long videos**, both now and in the future.
>
> DATE is indeed simple or may be 'engineering practice', but I think **simple but effective is the best**. Why do we need sophisticated but difficult-to-apply methods? Or, a **perfect story by decorating paper in formulas to highlight the theoretical aspects**, but is less effective than using a simpler way.
>
> Research is for applications not showing paper skills, and DATE has already been training-free applied to a real system since May. Whether or not more people see it, we have already made a value, and tried the best to submit to a public conference and enjoying the review process.
>
> Thank you again for your detailed suggestions and your professional peer review comments, this makes ICLR a great conference :)

---

### Official Review · Reviewer_zngB · 2025-11-09

**Soundness:** 4
**Presentation:** 3
**Contribution:** 3
**Rating:** 6
**Confidence:** 4

**Summary:**

This paper addresses the challenge of precise temporal reasoning and event localization in long video understanding for Multimodal Large Language Models (MLLMs). The authors argue that existing methods, which often use uniform frame sampling and implicit position encodings, fail to model absolute time and long-range dependencies effectively. The paper proposes a plug-and-play framework named DATE (Dynamic Absolute Time Enhancement) consisting of two main components. The first is the Temporal-Aware Similarity Sampling (TASS) strategy, which reframes sampling as a retrieval task: it first uses an LLM to convert a user's question into a descriptive caption, then uses CLIP similarity scores to perform a temporally-regularized greedy sampling (Algorithm 1) to select frames that are both semantically relevant and temporally diverse. The second component is the Timestamp Injection Mechanism (TIM), which explicitly interleaves textual timestamp tokens (e.g., "83s") with the corresponding video frame tokens, creating a direct and explicit temporal reference system for the MLLM. The authors demonstrate that DATE achieves state-of-the-art performance on several hour-long video benchmarks, with their 7B model variant outperforming even 72B models on some tasks.

**Strengths:**

1. The paper makes the key observation that prior semantic sampling methods (e.g., AKS ) use raw questions as input to CLIP. This is a mismatch, as CLIP was trained on descriptive captions. The TASS solution of first generating a caption from the question is a novel and clever fix.
2. The entire DATE framework is model-agnostic and requires no additional training. This is a significant practical advantage, lowering the barrier to adoption and making it easy to combine with future, stronger base models.
3. TASS's two-stage process (caption generation  + regularized greedy sampling ) directly addresses both the CLIP-mismatch and the problem of temporal redundancy in naive semantic sampling.
4. TIM's approach of interleaving explicit textual time_token while simplifying the MROPE to a simple sequential index  is an intresting way to decouple absolute time perception from relative order.

**Weaknesses:**

1. The TASS component requires calculating CLIP similarity for every frame (e.g., at 1 FPS)  and then running a greedy selection algorithm. The authors concede in Appendix D that this "leads to an inference time that grows approximately linearly with video length".
2. This linear cost is a significant practical drawback for the "hour-long" videos  that are the paper's primary target. A one-hour video at 1 FPS has 3,600 frames, which must all be encoded by CLIP before the MLLM even sees the input.
3. The paper fails to provide a quantitative analysis of this latency in the main text. While Table 3 compares TASS vs. AKS time, there is no plot or table showing total inference time (TASS + MLLM) vs. video duration (e.g., 10, 30, 60 minutes) to understand the practical impact of this bottleneck.
4. Figure 3  introduces T, H, and W notation for positional IDs without a clear textual explanation in Section 3.1 of what H and W (presumably spatial) represent or how they are indexed, which can be confusing on a first read. (I would suggest add a sentence to Section 3.1 briefly defining the T, H, and W notations used in Figure 3  to improve clarity.)
5. The TASS pipeline now critically depends on the quality of the caption generated by deepseek-v3. if the generated caption misinterprets the user query (e.g., ambiguous or overly generic phrasing), the CLIP similarity search can be misled, leading to suboptimal frame sampling.

**Questions:**

1. Could you provide a quantitative analysis of the TASS sampling latency? A plot of wall-clock time vs. video duration (e.g., 1 to 60 minutes) would be very helpful to understand the practical scalability of the method, as the linear cost  is a significant concern.
2. Regarding TASS+TIM Interaction, the TASS samples a sparse, non-uniform set of frames. TIM then applies simple sequential position IDs (1, 2, 3...) to these sampled frames, thereby ignoring the actual time gaps. This cleverly solves the large $\Delta k$ problem. However, does this mean the MLLM loses all sense of duration between sampled frames (since it only sees a relative order)? How does this impact reasoning about "how long" an event took?
3. The initial interval  is set to 20 seconds. While Figure 7 shows stability, this choice seems arbitrary. What happens if a video has a dense cluster of critical events all occurring within a 20-second window? Wouldn't this algorithm risk sampling only the first one and missing the rest?

---

> ### Author Response · Authors · 2025-11-16
> **Response to Official Review by Reviewer zngB**
>
> We sincerely appreciate your thoughtful reviews and advices. The responses of weaknesses and the question you concerned are listed below.
>
> # Weakness:
>
> ## W1:
>
> The **most time-consuming part is actually the video decoding**, more frames to sample may get better results, which reflects an **inherent trade-off between accuracy and efficiency** commonly observed in many video understanding tasks.
>
> CLIP similarity is not unique to TASS, but rather **a shared property of all similarity-based sampling approaches**.
>
> In future work, the computational efficiency could be improved by:
> 1. **Employing a lighter-weight vision model with lower output feature dimension**, reducing similarity computation cost while maintaining performance;
> 2. **Introducing a pre-screening step based on color or inter-frame differences** to filter out redundant frames before CLIP, thereby significantly lowering the overall cost.
>
> See more on **W3** for detailed computation time.
>
>
>
> ## W2:
>
> Yes, it is a drawback. It can increase the overall speed by reducing the sampling FPS, a trade-off between accuracy and efficiency
>
> However, for understanding very long videos, without using keyframe sampling, the model is essentially **guessing the answer** because the sampling density for long videos is **too low**, and key information will be **almost entirely lost**.
>
>
> ## W3:
>
> In Table 3, it is only the total sampling time on LVBench of 1,549 QA on CPU. Here, we provide a more detailed breakdown of the inference time for a single video at different frame counts on single A100-80G GPU. It shows **most of the time is spent on video decoding** (we use ```decord``` package), while the sampling time is relatively small.
>
>
> **Table I：The sampling time composition of a single video(in second)**
> | | | | | | |
> |-|-|-|-|-|-|
> |Total Frames |Video decode | CLIP encode   | Similarity compute | Sampling (TASS) |
> | | | | | | |
> |512 |3.15 | 0.56|0.31|0.009 |
> | 1024 | 5.19 |0.57|0.57|0.01 |
> | 2048 | 5.65 |0.64|1.09|0.01|
> |4096 | 9.78 |0.85|2.13|0.011 |
> | | | | | | |
>
>
> ## W4:
>
> Thank you for your suggestions, and we will improve it in the final version.
>
>
> ## W5:
>
> This is indeed a problem, and one that **all works that utilizing LLM face**. While misinterpretation is unavoidable, a brief rewrite (30 words) like the one in this paper **may not introduce very significant error**.
>
> Furthermore, we conducted related experiments, including the effects of rewriting using different language models, and the results are as follows:
>
> **Table II：TASS results with different caption generators on three benchmarks**
> | | | | | | | | |
> |-|-|-|-|-|-|-|-|
> | |Uniform (base)| Original query   | deepseek-v3 (used in this paper) | o4-mini | gemini-2.5-pro-preview | doubao-seed-1.6 | deepseek-R1 |
> | | | | | | | | |
> |Video-MME |55.44|55.33|57.33|56.78|57.11|57.11|57.67|
> | LongVideobench | 61.78   | 61.25   | 62.75   | 61.26 | 62.90    | 64.10        | 63.20  |
> | LVBench | 43.71   | 45.38  | 46.54   | 46.86 | 46.54   | 46.09  | 46.61   |
> | | | | | | | | |
>
>
> # Questions:
>
>
> ## Q1:
>
> Please refer to **W3**.
>
>
> ## Q2:
>
> In fact, TIM embeds the **absolute time** of the corresponding frame (e.g. 1.2s, 3.4s, 10.6s...), so the model can directly perceive the absolute time of each frame.
>
> After TASS sampling, the actual order is no longer the same duration. We use **TIM to time-stamp each frame**, which further enables the model to correctly perceive the time of frames.
>
> ## Q3:
>
> **The role of** $\delta_0$ is to control the expected sampling interval. For example, given a 2000s long video, when $\delta$ = 20, 10, or 5, each of the expectation of sampled frames is 100—meaning the sampling will be saturated at $\delta = 5$ seconds for 256 frames. For long videos, a 20/4 = 5s interval is relatively reasonable. So we choose 20s as the initial value. And the expectation is 5 means in a 20-second window it is expected to sample 4 frames.
>
> For shorter videos, there will be a finer-grained sampling, like $\delta$ = 2/1. Moreover, the initial value of $\delta_0$ can be reduced to achieve finer-grained sampling.

---

> > ### Comment · Reviewer_zngB · 2025-11-27
> >
> > I thank the authors for their detailed rebuttal, which has addressed most of my concerns. I agree that the proposed DATE workflow (combining TASS and TIM) presents a promising and practical solution for handling the challenges of hour-long video understanding.
> >
> > However, I remain concerned regarding the evaluation of the Timestamp Injection Mechanism (TIM) the same with other reviewer. Since the method is positioned as a "plug-and-play" training-free approach, stronger comparisons are essential to isolate the source of its performance gains. Currently, the only baseline provided for comparison is the "timestamp-in-prompt" method (Table 3). This baseline is relatively weak and does not sufficiently test the specific efficacy of the proposed interleaving strategy。
> >
> > To rigorously prove that the gains are derived specifically from your interleaving placement rather than just the presence of explicit time tokens, comparisons against other simple interleaving variants—such as different token placements, grouping timestamps at the start/end of segments, or using simple step markers—would have been valuable. Without such baselines, it remains unclear whether the interleaving structure itself is the critical factor.
> >
> > Despite the limited technical novelty, I recognize that the proposed method trys to convey a simple yet efficient idea for enhancing temporal awareness in MLLMs. Therefore, I will hold my current score.

---

> > > ### Author Response · Authors · 2025-11-28
> > >
> > > Thank you for your suggestion.
> > >
> > > Regarding the comparison between the timestamp embedding and other simple interleaving variants you mentioned, considering that the benchmark may not directly represent the true absolute time positioning capabilities, we visualized a simple **event location example at the end of the article (page 23, Appendix I)**. This videois 26 seconds long, using 64 frames as input, which Qwen2.5vl will merge into 32 segments.

---

### Author Response · Authors · 2025-11-28
**Confusion after a horizontal comparison**

Dear AC and Reviewers:

Thanks to the open-review feature, we noticed many submissions to this year's ICLR that focus on **video sampling (such as No.18064 and 6998)**, which also **employed the CLIP similarity sampling strategy** and **compared it with AKS**. Furthermore, No.6998 also **introduced LLM as an aid**, but **no one seems to care about the LLM illusion and computational cost** (because it's not meaningful; otherwise, all agents would cease to exist), and No.6998 may even have information leakage issues.

After comparing these publicly available results, we found that our **TASS significantly outperforms No.18064**. Based on our understanding of this area, we believe that our proposed method **has a greater advantage in simplicity and effectiveness, rather than narrative**.

**However**, both them were **highly recognized** with reviewers.

**It seems that the peer review process at conferences like ICLR is also a random lottery, which makes me confused.**

---

### Meta-Review · Area_Chair_J22L · 2026-01-07

**Summary:**

## Summary
 The paper proposes DATE, a plug-and-play framework (Dynamic Absolute Time Enhancement) for precise temporal reasoning and event localization in long video understanding.

## Discussion
All four reviewers share the same opinions about fact that the paper has limited technical novelty. While most of reviewers agree that the computational cost of DATE is a concern. AC agrees with reviewers' recommendation and thus recommends to reject the paper in its current form.

**Reviewer Concerns:**

* computational cost increases linearly to the video lenghth (zngB, EXdF, LxD8)
* depends on external components, i.e., deepseek-v3 (zngB, EXdF)
* limited technical novelty (zngB, f994, EXdF, LxD8)

**Reviewer Scores:**

The paper recieves the scores of 6, 4, 4, 4 from 4 different reviewers and most of reviewers state that their opinion about the paper unchanged after rebuttal.

---

### Decision · Program_Chairs · 2026-01-26

Reject